# Distilling the Thought, Watermarking the Answer: A Principle Semantic Guided Watermark for Large Reasoning Models

**Shuliang Liu**[1,2]**, Xingyu Li**[1]**, Hongyi Liu**[1]**, Dong Fang**[3*]**,**
**Bingchen Duan**[1,2]**, Qi Zheng**[1,2]**, Lingfeng Su**[3]**, Xuming Hu**[1,2*]
[1] The Hong Kong University of Science and Technology (Guangzhou)
[2] The Hong Kong University of Science and Technology
[3] LIGHTSPEED
shulianglyo@gmail.com, df572@outlook.com
slf568925701@gmail.com, xuminghu@hkust-gz.edu.cn

## Abstract

Reasoning Large Language Models (RLLMs) excelling in complex tasks present unique challenges for digital watermarking, as existing methods often disrupt logical coherence or incur high computational costs. Token-based watermarking techniques can corrupt the reasoning flow by applying pseudo-random biases, while semantic-aware approaches improve quality but introduce significant latency or require auxiliary models. This paper introduces **ReasonMark**, a novel watermarking framework specifically designed for reasoning-intensive LLMs. Our approach decouples generation into an undisturbed Thinking Phase and a watermarked Answering Phase. We propose a Criticality Score to identify semantically pivotal tokens from the reasoning trace, which are distilled into a Principal Semantic Vector (PSV). The PSV then guides a semantically-adaptive mechanism that modulates watermark strength based on token-PSV alignment, ensuring robustness without compromising logical integrity. Extensive experiments show ReasonMark surpasses state-of-the-art methods by reducing text Perplexity by 0.35, increasing translation BLEU score by 0.164, and raising mathematical accuracy by 0.67 points. These advancements are achieved alongside a 0.34% higher watermark detection AUC and stronger robustness to attacks, all with a negligible increase in latency. This work enables the traceable and trustworthy deployment of reasoning LLMs in real-world applications. The source code is available here.

## 1 Introduction

Large Language Models (LLMs) have demonstrated remarkable advancements in recent years, achieving state-of-the-art performance across a multitude of domains including information retrieval (Labruna et al., 2024; Jin et al., 2025; Hu et al., 2022; Zhang et al., 2025a), medical diagnosis (Zhou et al., 2024), financial analysis (Li et al., 2023b; Lopez-Lira et al., 2025), legal assistance (Kuk & Harasta, 2025; Fei et al., 2023), academic research (Liao et al., 2024; Naveed et al., 2023), and speech processing (Hei et al., 2025). More recently, a new wave of models, exemplified by systems like ChatGPT-4o (Jaech et al., 2024) and DeepSeek-V2 (Guo et al., 2025), have showcased superior capabilities in complex reasoning tasks such as mathematical problem-solving, strategic planning, code generation, and scientific discovery (Guo et al., 2025; Wei et al., 2025; Wen et al., 2025; Liu et al., 2026; Ling et al., 2025b). These reasoning-intensive LLMs often employ distinct training paradigms and inference mechanisms, such as internal monologues or chain-of-thought (CoT) prompting (Wei et al., 2022), which differentiate them significantly from their predecessors.

The burgeoning capabilities and widespread adoption of LLMs, particularly those adept at reasoning, necessitate robust mechanisms for ensuring content authenticity, traceability, and intellectual property protection. Digital watermarking (Kirchenbauer et al., 2023) has emerged as a promising technique

---

[*]Corresponding authors: Dong Fang and Xuming Hu.

to invisibly embed identifiable signals within model-generated text, thereby enabling source tracking and mitigating misuse (Abdelnabi & Fritz, 2021; Chang et al., 2024; Hou et al., 2023). However, existing watermarking algorithms, largely developed for general-purpose LLMs, face significant challenges when applied to reasoning-based models. For instance, methods like KGW (Kirchenbauer et al., 2023), which rely on pseudo-random vocabulary partitioning, can inadvertently disrupt the logical consistency of the model's internal reasoning—the *thinking phase*—thereby compromising the coherence and accuracy of the final answer. Other approaches that focus on preserving text quality, such as unbiased sampling techniques (Hu et al., 2023), often do so at the cost of detection efficiency. Conversely, methods like EWD (Lu et al., 2024) and SWEET (Lee et al., 2023), while achieving higher detection rates, may introduce perceptible artifacts that degrade text quality. More sophisticated strategies like WaterMax (Giboulot & Furon, 2024), which perform multiple generation runs to find optimally watermarked outputs, achieve a better balance but incur substantial computational overhead and increased inference latency. This persistent trade-off among text quality, watermark detectability, and computational efficiency has hindered the practical deployment of watermarking in many real-world applications (Liu et al., 2024; Huang et al., 2025).

To address these challenges, we introduce **ReasonMark**, a novel watermarking framework specifically designed for large reasoning models, centered on the principle of *Distilling the Thought, Watermarking the Answer*, as illustrated in Fig. 1. Our approach decouples the generation process into two distinct stages: an undisturbed internal Thinking Phase, where the model performs its reasoning, and a subsequent Answering Phase, where the final response is generated. The core innovation lies in preserving the integrity of the thinking phase entirely. We analyze it to identify a set of **Criticality Tokens (CTs)** that encapsulate the most salient semantic anchors of the reasoning process. These tokens are then distilled into a continuous vector representation, the **Principal Semantic Vector (PSV)**, which serves as a dynamic semantic compass for the answering phase. The PSV guides a semantically-adaptive watermarking mechanism, where the watermark strength applied to candidate tokens is modulated by their alignment with the model's established reasoning trajectory. By aligning the watermark with the model's own logical flow, ReasonMark can embed a robust and detectable signal without disrupting coherence or accuracy. This effectively resolves the debilitating trade-off between watermark strength and semantic integrity, all while avoiding the additional inference latency common in other semantic-aware techniques. Our main contributions are threefold:

- We propose a novel two-phase watermarking framework that decouples a model's internal reasoning from its final answer generation. This is the first approach specifically designed to protect the outputs of RLLMs without corrupting their logical integrity.

- We design a principled method to distill the semantic essence of the model's reasoning process, involving a Criticality Score to identify key tokens and their subsequent transformation into a PSV that provides a continuous, directional guide for watermarking.

- Extensive experiments show ReasonMark surpasses baselines by reducing text PPL by 0.35, increasing translation BLEU score by 0.164, and raising mathematical accuracy by 0.67 points, while also improving detection AUC by 0.34% with negligible latency.

## 2   PRELIMINARY

In this section, we introduce token-based and semantic-based watermarking methods, outlining their respective strengths and limitations. We then formalize the framework of our proposed algorithm and define its key concepts.

**Related Work.**   Existing LLM watermarking research is primarily divided into token-based and semantic-based approaches. Token-based methods, such as the seminal work by Kirchenbauer et al. (2023), partition the vocabulary and apply a statistical bias during generation (Hu et al., 2023). While effective for detection, their pseudo-random nature can disrupt the logical flow and semantic consistency crucial for reasoning tasks (Yoo et al., 2024; Chang et al., 2024). Adaptive watermark strength methods like (Wang et al., 2025b) balance the effectiveness-quality trade-off at the list level but ignore the intra-list importance of individual tokens. Conversely, semantic-based methods operate in the embedding space to improve text quality and robustness against paraphrasing (Ren et al., 2023; Hou et al., 2023; Huo et al., 2025). However, they often introduce significant computational overhead by requiring auxiliary models or architectural modifications (Baldassini et al., 2024), and

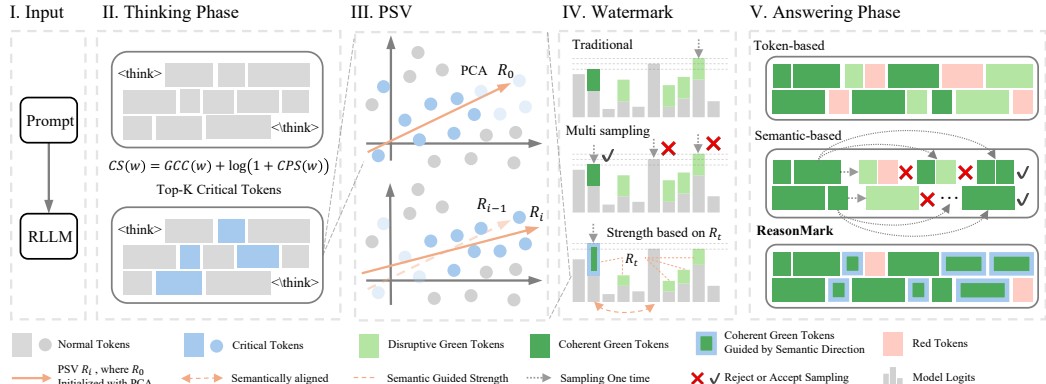

Figure 1: ReasonMark identifies top-K critical tokens during the thinking phase (II.) and uses PCA (III.) to establish an initial Principal Semantic Vector (PSV). This semantic vector then guides the watermarking process (IV.) by dynamically adjusting the logits to favor semantically coherent green tokens and penalize disruptive ones.This enables the efficient generation of a semantically coherent watermarked sequence with a high proportion of green tokens (V.) by sampling only one time.

are not specifically tailored to preserve the step-by-step integrity of complex reasoning (Dasgupta et al., 2024). Our PSV-guided watermarking framework bridges this gap by distilling the reasoning phase into a continuous Principal Semantic Vector, which dynamically modulates watermark strength based on token-PSV alignment. This approach ensures the watermark is semantically coherent, thus preserving logical integrity, a key weakness of token-based methods, while maintaining high detection efficiency and avoiding the extra inference costs typical of many semantic-aware techniques (Baldassini et al., 2024). A more comprehensive review of related work is available in Appendix B.

**Phase Segmentation.** Let $T = \{t_1, t_2, ..., t_S\}$ denote the full sequence of $S$ tokens generated by a language model. We partition $T$ into two distinct phases. The **Thinking Phase**: $T_{\text{think}} = \{t_i\}_{i=1}^N = \{t_1, t_2, ..., t_N\}$, comprising the model's internal chain-of-thought or reasoning steps. The **Answering Phase**: $T_{\text{answer}} = \{t_i\}_{N+1}^S = \{t_{N+1}, ..., t_S\}$, representing the final response intended for the user. The delineation point $k$ is identified via a *Marker-Based Separation Algorithm* (Guo et al., 2025) that detects structural delimiters (e.g., <think>, <\think>), as illustrated in Fig 1.II.

**Definition 2.1** (Semantic Guidance via Principal Semantic Vector). *Given the thinking phase $T_{think}$, the sequence of probability distributions over the vocabulary $\mathcal{V}$ during thinking phase $\{P_i\}_{i=1}^N$, and the previously generated answer tokens $\{t_{N+1}, ..., t_i\}$, we define:*

$$\mathcal{R}_0 = f_\eta(\{t_i\}_{i=1}^N, \{P_i\}_{i=1}^N), \quad \delta_{i,t_i} = g_\sigma(\mathcal{R}_{i-1}, t_i), \quad \mathcal{R}_i = f_\mu(\mathcal{R}_{i-1}, t_i) \tag{1}$$

Here, the initial PSV $\mathcal{R}_0$ captures the reasoning trajectory, while $\delta_{i,t_i}$ provides watermark guidance to token $t_i$ at step $i$. The PSV $\mathcal{R}_i$ updates dynamically to reflect the evolving semantic context, ensuring watermark strength aligns with the model's logical flow throughout answer generation. The design of $f_\eta$ (Fig. 1.II. to III.) poses the primary challenge; thus, the rest of this section details its design, while the implementations of $g_\sigma$ (Fig. 1.IV.) and $f_\mu$ (Fig. 1.III.) are deferred to Section 3.2.

**Principle Semantic Vector Construction.** The foundation for constructing PSV rests upon identifying a curated set of **Critical Tokens (CTs)** within the thinking phase $T_{\text{think}}$, inspired by (Liu et al., 2025b; Zheng et al., 2026). These CTs are hypothesized to encapsulate the most salient semantic anchors of the reasoning process and thus provide the essential raw material for deriving the initial PSV. We then formalize the notion that a token's criticality is a function of both its influence on the generative trajectory and its ability to resolve uncertainty. This principle is encapsulated in the following theorem, which defines an optimal set of CTs.

**Theorem 2.2** (Optimal Representation of Critical Tokens). *The optimal set of Critical Tokens, denoted $\mathcal{C}^* \subseteq \mathcal{V}$, is the set that maximizes a joint measure of causal influence and competitive significance, subject to a constraint $|\mathcal{C}| \leq K$ on its size:*

$$\mathcal{C}^* = \arg \max_{\mathcal{C} \subseteq \mathcal{V}, |\mathcal{C}| \leq K} \sum_{w \in \mathcal{C}} [D_{causal}(w\|\theta) + \omega \cdot \mathbb{E}_{j>i} [\Delta S_{i \to j}(w)]] \tag{2}$$

*where $K$ is the maximum desired number of Critical Tokens, $\theta$ represents model parameters, and $\omega$ balances the two measures. The Causal Divergence $D_{causal}(w\|\theta)$ is:*

$$D_{causal}(w\|\theta) = \sum_{i=1}^{N} \lambda_i \cdot \|\nabla_\theta \mathbb{E}_{w\prime \sim P_i}[Sim(w, w\prime)]\|^2 \tag{3}$$

*and the Competitive Entropy Reduction $\Delta S_{i \to j}(w)$ is:*

$$\Delta S_{i \to j}(w) = S(P_j) - S(P_j | w \in Top_k(P_i)) \tag{4}$$

Further explanation can be seen at Appendix C. While Theorem 2.2 provides a principled foundation, the direct computation of the causal divergence term, which requires evaluating gradients with respect to all model parameters $\theta$, is computationally prohibitive for large models.

## 3 METHODOLOGY

### 3.1 ALGORITHMIC REALIZATION OF PSV CONSTRUCTION $f_\eta$

This construction is primarily realized through Critical Tokens (CTs). Consequently, this section is organized into two main parts: the first details the method for identifying CTs, and the second explains how these tokens are utilized to construct PSV, corresponding to the function $f_\eta$ in Eq. 1.

#### 3.1.1 CRITICALITY SCORE

Translating Theorem 2.2 into a practical algorithm, we devise a Criticality Score for each word $w \in \mathcal{V}$. The proof of its validity is discussed in detail in Appendix D. This score is a composite measure reflecting both the global causal influence and the local competitive persistence of $w$.

**Global Causal Contribution (GCC).** This component aims to approximate $D_{\text{causal}}(w\|\theta)$ (Eq. 3) by quantifying a word $w$'s capacity to indirectly shape the reasoning trajectory through sustained high probability in causally interconnected steps. The GCC is formulated as:

$$\text{GCC}(w) = \sum_{i=1}^{N} \left[ P_i(w) \cdot \lambda_i \cdot \sum_{j=i+1}^{M} \alpha_{i \to j} \cdot P_j(w) \right] \tag{5}$$

The weight $\lambda_i = \text{JS}(P_i\|P_{i-1})$ captures the magnitude of change in the models predictive distribution from step $i-1$ to $i$. A large JS divergence signals a critical juncture in the reasoning process, amplifying the contribution of words prominent at such points. The term $\alpha_{i \to j} = \frac{\text{Sim}(P_i, P_j)}{\sum_{k\prime=1}^{N} \text{Sim}(P_i, P_{k\prime})}$ represents the normalized semantic influence of the distributional state at step $i$ on that of step $j$. Here, $\text{Sim}(P_i, P_j) = \frac{\mathbf{P}_i \cdot \mathbf{P}_j}{\|\mathbf{P}_i\|\|\mathbf{P}_j\|}$ is the cosine similarity between the vector representations of probability distributions $P_i$ and $P_j$. This factor ensures that the influence of early critical steps is appropriately propagated and weighted in assessing a words contribution to later stages of reasoning.

**Competitive Persistence Scoring (CPS).** This component approximates $\mathbb{E}_{j>i}[\Delta S_{i \to j}(w)]$ (from Eq. 4) by rewarding words that not only feature prominently in competitive generation contexts but also maintain this prominence over subsequent steps. The CPS for a word $w$ is calculated as:

$$\text{CPS}(w) = \sum_{i=1}^{N} \left[ S(t_i)^{-1} \cdot (1 - \Delta_i(w)) \cdot \sum_{j=i+1}^{M} \mathbb{I}(w \in \text{top}_k(P_j)) \right] \tag{6}$$

The term $S(t_i)^{-1} = (-\log P_i(t_i))^{-1}$ inversely weights the contribution by the surprisal of the token $t_i$ actually generated at step $i$. This rewards contexts where the model makes a high-confidence choice, suggesting that such choices are more deliberate and impactful. The core of this reward lies in $\Delta_i(w)$, which measures the competitive pressure surrounding $w$ at step $i$:

$$\Delta_i(w) = \begin{cases} |L_i(w) - \max_{v \neq w} L_i(v)|, & \text{if } w = t_i \text{ (i.e., } w \text{ was selected)} \\ |L_i(w) - L_i(t_i)|, & \text{if } w \in \text{top}_k(L_i) \text{ and } w \neq t_i \text{ (i.e., } w \text{ was a close competitor)} \\ 1, & \text{otherwise (not competitive)} \end{cases}$$

$$\tag{7}$$

When $w$ is the selected token $t_i$, $\Delta_i(w)$ is its logit margin over the strongest competitor. If $w$ was a top-k candidate but not selected, $\Delta_i(w)$ is its logit difference from the winner $t_i$. A smaller $\Delta_i(w)$ indicates more intense competition. The reward thus assigns higher rewards to tokens that emerge from, or are central to, highly contested selection points.

$\sum_{j=i+1}^{M} \mathbb{I}(w \in \text{top}_k(P_j))$ counts the number of times $w$ appears among the top-$k$ probability candidates in the $M - i$ steps immediately following step $i$. This serves as empirical validation of $w$'s enduring relevance and high-frequency consideration throughout the local reasoning window, reinforcing its status as a critical element.

**Consolidated Criticality Score (CS).** The final score synergistically combines these two aspects to provide a holistic measure of a token's importance.

$$\text{CS}(w) = \text{GCC}(w) \cdot \log(1 + \text{CPS}(w)) \tag{8}$$

The set of Critical Tokens $\mathcal{C}'$, is then formed by selecting the $K$ tokens with the highest CS values, as Fig 1.II illustrated, providing the semantic anchors for the next stage of our methodology. The case study in Appendix H examines the distribution of normalized CS for CTs on different datasets, revealing their correspondence with the input and output of the model.

### 3.1.2 FROM CRITICAL TOKENS TO PRINCIPAL SEMANTIC VECTOR

While the discrete set $\mathcal{C}'$ identifies key semantic anchors, it falls short of capturing the holistic, relational logic inherent in complex reasoning. To overcome this limitation, we transform this discrete set of tokens into a continuous vector representation, the **PSV**, that encapsulates the dominant semantic direction of the entire thinking phase.

Let $E(\cdot)$ be the model's token embedding function. We first construct an embedding matrix $H \in \mathbb{R}^{K \times d}$ by stacking the embeddings of the $K$ identified Critical Tokens from $\mathcal{C}'$, where $d$ is the embedding dimension.

$$H = [E(w_1), E(w_2), ..., E(w_K)]^T, \quad \forall w_i \in \mathcal{C}' \tag{9}$$

We then apply Principal Component Analysis (PCA) to $H$. The first principal component $\mathbf{v}_1$, represents the direction of maximum variance within the embeddings of the most critical tokens. This direction captures their most significant shared semantic properties and reflects the primary axis of the model's reasoning. We define the initial PSV $\mathcal{R}_0$ as this first principal component:

$$\mathcal{R}_0 = \mathbf{v}_1 = \text{PCA}_1(H) \tag{10}$$

This initial PSV $\mathcal{R}_0$, described in Definition 2.1 and Fig. 1.III, acts as a global semantic compass, providing a stable, overarching directional guide for the subsequent answering phase.

### 3.2 SEMANTICALLY-ADAPTIVE WATERMARK EMBEDDING $g_\sigma$ AND $f_\mu$

Our framework departs from conventional methods that employ a fixed watermark strength (Kirchenbauer et al., 2023), inspired by Wang et al. (2025b). Instead, we introduce a semantically-adaptive mechanism where the watermark's intensity is dynamically modulated based on the alignment of candidate tokens with the current PSV. This allows for a strong watermark signature on semantically coherent tokens while minimizing interference with the model's natural generation process.

**Dynamic Watermark Strength.** At each generation step $i$ in the answer phase, we partition the vocabulary $\mathcal{V}$ into a green list $\mathcal{V}_g$ and a red list $\mathcal{V}_r$ based on a standard cryptographic hash of the previous token, following Kirchenbauer et al. (2023). However, instead of applying a fixed bonus $\delta$ to the logits of all green-list tokens, we compute a token-specific bonus $\delta_{i,w}$ for each candidate token $w \in \mathcal{V}_g$. This bonus is proportional to the token's semantic relevance to the current PSV $\mathcal{R}_{i-1}$:

$$s_{w,i} = \frac{E(w) \cdot \mathcal{R}_i}{\|E(w)\| \|\mathcal{R}_i\|}, \quad \delta_{i,w} = \delta_0 + \delta_\lambda \cdot s_{w,i-1} \tag{11}$$

where $s_{w,i-1}$ is the cosine similarity between the embedding of token $v$ and the PSV $\mathcal{R}_{i-1}$. $\delta_0$ is a base watermark strength, and $\delta_\lambda$ is a scaling factor that controls the sensitivity to semantic alignment.

Table 1: Experimental results on C4, WMT16-DE-EN, AIME, and GSM8K datasets for Qwen3-32B and Deepseek-R1 32B models. We report Perplexity (PPL)(↓) for text quality, BLEU and mACC(↑), short for math ACC, for task performance, and AUC (↑) for watermark detection. The best result among watermarking methods for each metric is in **bold**.

| | C4 | | | | WMT | | | | AIME | | | | GSM8K | | | |
| | Qwen3 | | Deepseek | | Qwen3 | | Deepseek | | Qwen3 | | Deepseek | | Qwen3 | | Deepseek | |
| **Method** | PPL | AUC | PPL | AUC | BLEU | AUC | BLEU | AUC | mACC | AUC | mACC | AUC | mACC | AUC | mACC | AUC |
| No Watermark | 10.55 | - | 10.82 | - | 7.851 | - | 7.622 | - | 70.03 | - | 71.52 | - | 94.01 | - | 95.21 | - |
| KGW | 12.15 | 98.78 | 12.52 | 98.55 | 7.351 | 82.36 | 7.185 | 81.95 | 69.23 | 98.16 | 70.67 | 98.43 | 92.98 | 94.11 | 94.16 | 94.57 |
| UPV | 11.41 | 97.01 | 11.62 | 97.15 | 7.493 | 82.75 | 7.288 | 82.50 | 63.04 | 86.94 | 64.23 | 87.46 | 92.51 | 81.92 | 93.67 | 82.45 |
| Unigram | 10.66 | 97.10 | 10.97 | 96.90 | 8.441 | 83.26 | 8.157 | 83.01 | 56.02 | 93.27 | 57.36 | 93.62 | 93.88 | 82.53 | 94.94 | 83.07 |
| Unbiased | 11.52 | 93.06 | 11.92 | 92.58 | 7.516 | 84.50 | 7.305 | 84.21 | 65.51 | 85.73 | 66.83 | 86.26 | 92.17 | 80.52 | 93.21 | 81.04 |
| SynthID | 12.69 | 87.61 | 13.22 | 87.11 | 6.953 | 78.15 | 6.781 | 77.86 | 52.34 | 83.12 | 53.65 | 83.67 | 90.53 | 74.24 | 91.46 | 74.88 |
| SWEET | 12.46 | 97.27 | 12.62 | 97.20 | 7.209 | 85.10 | 7.016 | 84.88 | 66.01 | 99.86 | 67.28 | 99.89 | 93.74 | 92.51 | 94.82 | 93.13 |
| EWD | 11.89 | 99.22 | 12.12 | 99.18 | 7.413 | 86.80 | 7.228 | 86.45 | 69.52 | 99.91 | 71.04 | 99.94 | 93.67 | 95.82 | 94.76 | 96.41 |
| WatMe | 11.27 | 98.53 | 11.67 | 98.60 | 8.038 | 86.93 | 7.893 | **86.55** | 67.03 | 88.11 | 68.46 | 88.53 | 93.82 | 84.25 | 94.87 | 84.74 |
| MorphMark | 11.01 | 94.16 | 11.22 | 94.55 | 9.752 | 76.08 | 9.463 | 75.82 | 68.74 | 88.31 | 70.17 | 88.79 | 93.52 | 76.76 | 94.63 | 77.15 |
| SemStamp | 11.42 | 97.85 | 11.73 | 97.65 | 7.912 | 85.20 | 7.682 | 84.80 | 68.90 | 98.95 | 70.31 | 99.15 | 93.05 | 94.80 | 94.28 | 95.38 |
| k-SemStamp | 11.22 | 98.10 | 11.51 | 97.90 | 8.123 | 85.50 | 7.886 | 85.15 | 69.15 | 99.25 | 70.55 | 99.35 | 93.25 | 95.10 | 94.45 | 95.65 |
| SimMark | 11.18 | 97.95 | 11.46 | 97.75 | 8.191 | 85.40 | 7.954 | 85.00 | 69.05 | 99.10 | 70.48 | 99.23 | 93.18 | 94.95 | 94.39 | 95.52 |
| **ReasonMark** | **10.31** | **99.31** | **10.54** | **99.52** | **9.916** | **87.25** | **9.653** | 85.10 | **69.86** | **99.95** | **71.34** | **99.98** | **93.96** | **95.94** | **95.14** | **96.56** |

The logit for a green-list token $w$ is then modified as $L_i(w) \leftarrow L_i(w) + \delta_{i,w}$. This ensures that green-list tokens that are highly aligned with the intended reasoning trajectory receive a stronger watermark, reinforcing logical consistency. If a highly probable, contextually appropriate token falls into the red list, the relatively lower bonuses on green-list alternatives prevent significant quality degradation. The effect of these two parameters on model performance is analyzed in Section 4.5.

**Dynamic PSV Update.** The PSV is not static; it evolves with the generation of the answer to act as a semantic compass, tracking the local semantic context. After a token $t_i$ is generated at step $i$, we update the PSV using an exponential moving average:

$$\mathcal{R}_i = (1 - \beta_i)\mathcal{R}_{i-1} + \beta_i E(t_i), \quad where \ \beta_i = \beta_{\text{base}} \cdot s_{t_i, i-1} \tag{12}$$

The update rate $\beta_i$ is itself adaptive, depending on the semantic contribution of the newly generated token, where $\beta_{\text{base}} \in [0, 1]$ is a small base learning rate, which is also analyzed in Section 4.5. This mechanism ensures that the PSV gradually incorporates the semantic content of the unfolding answer, allowing for smooth topical transitions while remaining anchored to the initial reasoning established in $T_{\text{think}}$.

**Watermark Detection.** A significant advantage of our approach is that the detection process requires no modification to standard procedures in Kirchenbauer et al. (2023). Despite the dynamic nature of the watermark embedding, detection remains stateless and does not require access to the PSV or the original prompt. It is performed using the same statistical z-test as in KGW by checking for a statistically significant bias towards green-list tokens in the generated text. The performance gain of the algorithm is attributed to its ability to identify a larger set of valid green tokens from the candidate list at each step $i$, thereby reducing the number of red tokens.

# 4 EXPERIMENTS

## 4.1 EXPERIMENTAL SETTINGS

**Datasets and Prompts.** We evaluate our method on datasets spanning both text generation and reasoning tasks. (1) Our evaluation of text generation encompasses two primary tasks. The first is text completion, for which we adopt the C4 dataset (Raffel et al., 2023), which is widely used in prior watermarking studies. The first 30 tokens of each sample are taken as prompts, and the model generates the continuation. The second is machine translation, and we use the WMT16 German–English dataset (Bojar et al., 2016), where the task is to translate German sentences into English. (2) For reasoning, we employ the AIME (Veeraboina, 2023) and GSM8K (Cobbe et al., 2021) mathematical benchmark, which provides standardized solutions, enabling a rigorous evaluation

of watermarking in tasks where correctness can be objectively assessed. The prompts used in our experiments are detailed in Appendix E.1.

**Models and Baselines.** We conduct experiments with Qwen3-32B (Yang et al., 2025) and DeepSeek-R1-Distill-Qwen-32B (abbreviated as DeepSeek in the subsequent section) (Guo et al., 2025) models. To ensure a comprehensive comparison, we benchmark our method against a range of representative watermarking algorithms, including token-based methods KGW (Kirchenbauer et al., 2023), UPV (Liu et al., 2023a), Unigram (Zhao et al., 2023), Unbiased (Hu et al., 2023), SWEET (Lee et al., 2023), EWD (Lu et al., 2024), WatMe (Chen et al., 2024), and MorphMark (Wang et al., 2025b), as well as semantic-based approaches SemStamp (Hou et al., 2023), k-SemStamp (Hou et al., 2024), and SimMark (Dabiriaghdam & Wang, 2025). Implementations are facilitated by the MarkLLM (Pan et al., 2024) repository. Evaluation Metrics are detailed discussed in Appendix E.2.

**Hyperparameters.** For methods requiring a $\delta$ parameter (e.g., KGW, Unigram), we set $\delta = 2$ by default. And we set $\delta_0 = 1.5, \delta_\lambda = 3.0$ in Eq. 11, which is analysed in Sec 4.5. For text generation tasks, we apply repetition penalties to reduce duplicate outputs, including the `no_repeat_ngram_size` constraint. For mathematical reasoning tasks, however, we refrain from imposing such penalties, as preliminary experiments showed that these constraints significantly reduce problem-solving accuracy, regardless of watermarking.

## 4.2 MAIN RESULTS

As presented in Table 1, ReasonMark demonstrates a superior balance between output quality, task performance, and watermark detectability across all evaluated datasets and models, consistently outperforming existing state-of-the-art methods. A comparative analysis of inference latency, averaged over multiple executions, is detailed in Appendix F.2. The results demonstrate that our method's computational overhead is highly competitive. Furthermore, we provide case studies in Appendix H and visualization of PSV and CTs in Appendix F.4, examining the identified critical tokens and the embedded watermark to offer qualitative insights into the efficacy of our algorithm.

On text generation tasks, our method achieves the lowest perplexity (PPL) on the C4 dataset (10.31 for Qwen3-32B and 10.54 for Deepseek-R1 32B), indicating the highest text quality that is nearly on par with non-watermarked text. For machine translation on WMT16-DE-EN, ReasonMark obtains the highest BLEU scores among all watermarking techniques (9.916 and 9.653), showcasing its ability to preserve translation fidelity. A more detailed breakdown of the results for various translation metrics can be found in Appendix F.1.

Crucially, in reasoning-intensive benchmarks, our approach excels at maintaining logical integrity. On both AIME and GSM8K, ReasonMark achieves the highest mathematical accuracy (mACC), closely matching or even slightly exceeding the baseline performance without a watermark, while other methods often lead to a noticeable degradation in performance. For instance, on the AIME dataset with the Deepseek model, our method scores 71.34 in mACC, surpassing all other watermarking techniques and nearing the 71.52 of the non-watermarked baseline.

Across all these tasks, ReasonMark consistently delivers the highest or near-highest detection rates, with an AUC of 99.31 and 99.52 on C4, and over 99.9 on the AIME reasoning task. This empirically validates that our framework effectively resolves the trade-off between semantic integrity and watermark robustness, preserving the performance of reasoning LLMs while embedding a strong, detectable signal.

## 4.3 ATTACK ROBUSTNESS ANALYSIS

In our robustness experiments, we evaluated multiple watermarking algorithms under two attack settings, A1 and A2 (Lau et al., 2024) on C4 dataset using Qwen3 model. Attack type A1 applies random word-level perturbations, including insertions, deletions, and synonym substitutions, with 30 of the tokens modified. Attack type A2 consists of semantic-level transformations via translation and paraphrasing, which are implemented through calls to the DeepSeek-V3 API. For the translation attack, each text is translated into Chinese and subsequently back into English.

As shown in Fig 2, ReasonMark demonstrates superior robustness against both word-level and semantic-level attacks. Achieving a near-perfect unattacked AUC of 99.31, it maintains high detectability above 93.5 under word deletion, insertion, and synonym replacement. Crucially, it shows

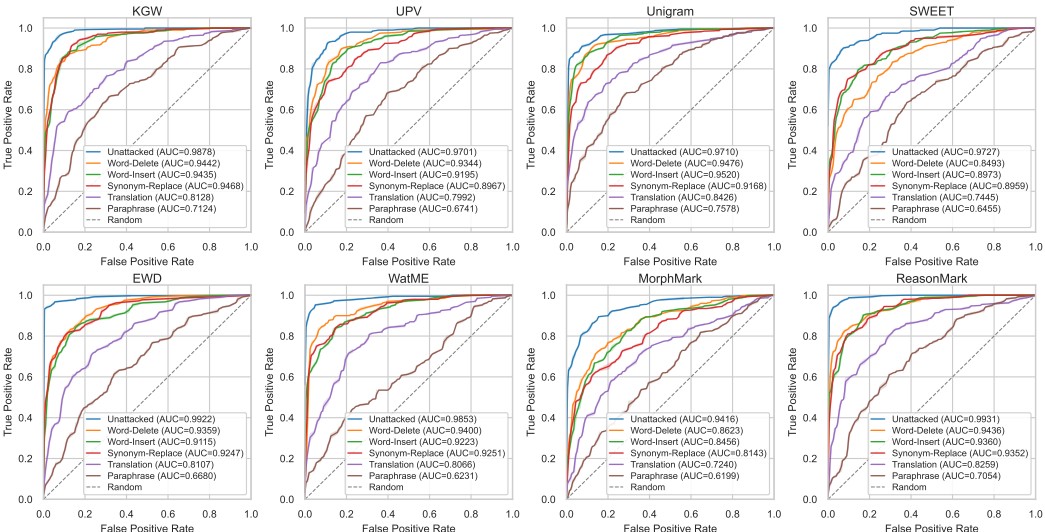

Figure 2: ROC curves under different attack methods for various watermarking approaches.

strong resilience to semantic attacks, retaining a high AUC of 82.58 against translation and 70.54 against paraphrasing. This resilience stems from our core principle of embedding the watermark in alignment with the model's reasoning, captured by the PSV. By tying the watermark to the core semantic meaning rather than the syntactic structure, ReasonMark ensures persistence against such modifications, validating the efficacy of our semantically-grounded approach in adversarial settings. More experiments regarding machine translation and detailed attack robustness evaluation can be found in Appendix F.1 and Appendix F.3, respectively.

## 4.4 ABLATION STUDY

To validate our core components, we conducted an ablation study in Table 2 with Qwen3 on C4 dataset. We tested variants by replacing our Critical Token (CT) selection with random sampling (**w/o CTs**), and by individually removing the Global Causal Contribution (**w/o GCC**) and Competitive Persistence Scoring (**w/o CPS**) modules to evaluate the effect of each components.

The results confirm that all components are essential. The full ReasonMark model achieves the best performance, with the lowest perplexity (10.3080) and a high AUC (0.9931). The most significant performance drop occurred in the **w/o CTs** variant, where PPL increased to 12.8801. This demonstrates that our principled, semantic-based token selection is critical for maintaining text quality, as random tokens fail to provide coherent guidance for the watermark. Furthermore, removing the

Table 2: Ablation study on the C4 dataset.

| Method / Variant | PPL | AUC |
|---|---|---|
| No Watermark | 10.5488 | - |
| KGW | 12.1503 | 98.78 |
| WatMe | 11.2734 | 98.53 |
| SimMark | 11.1783 | 97.95 |
| **ReasonMark** | **10.3080** | **99.31** |
| w/o CTs | 12.8801 | 99.21 |
| w/o GCC | 11.1510 | 99.11 |
| w/o CPS | 11.0597 | 98.69 |

GCC or CPS modules also degrades performance. The absence of GCC (**w/o GCC**) primarily impacts text quality (PPL increases to 11.1510), while removing CPS (**w/o CPS**) leads to a more noticeable drop in watermark detectability (AUC falls to 98.69). However, it is noteworthy that even without these individual modules, the **w/o GCC** and **w/o CPS** variants still maintain highly competitive overall performance, matching or even surpassing the majority of the baseline methods (such as KGW, WatMe, and SimMark) in both text quality and detectability. This shows that GCC is vital for semantic coherence, and CPS is crucial for embedding a robust watermark. In conclusion, the components are synergistic and indispensable for achieving the optimal balance between text quality and detection robustness.

## 4.5 HYPERPARAMETER SENSITIVITY ANALYSIS

**$\beta_0$ and Top-k Parameter Analysis.** To understand the impact of key hyperparameters on our method's performance, we conduct a sensitivity analysis for the PSV update rate $\beta_0$ and the top-k sampling value. Figure 3 illustrates how text quality, measured by perplexity, varies with these parameters. The analysis reveals that ReasonMark is robust, showing stable performance across a wide range of values for both hyperparameters.

For the PSV update rate $\beta_0$, perplexity follows a U-shaped curve, starting at 11.1 for a value of $10^{-3}$, reaching its minimum of approximately 10.3 around 0.1, and then increasing again. Similarly, the top-k parameter shows

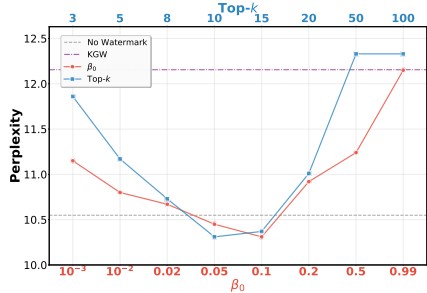

Figure 3: Visualization of $\beta_0$ and top-k.

that perplexity is highest at a small k of 3, drops to its lowest point around k=10, and then gradually rises as k increases to 100. Notably, the perplexity of ReasonMark consistently remains well below the KGW baseline across all tested settings, underscoring a persistent advantage in text quality. Critically, the results highlight that with careful tuning, our method's performance can even surpass the non-watermarked baseline. The optimal configuration, with a $\beta_0$ value in the range of 0.01 to 0.1 and a top-k value between 10 and 50, yields a perplexity score that is superior to that of the original, non-watermarked text. This demonstrates that ReasonMark not only embeds a robust watermark but can also enhance text fluency.

**$\delta_0$ and $\delta_\lambda$ Interaction Analysis.** To maintain the overall watermark strength approximately consistent with other methods in our hyperparameter settings, specifically $\delta = 2$, as outlined in Section 2, and considering the formulation in Eq. 7, we set $\delta_0$ within a range of 1 to 2. This ensures that the overall watermark strength does not deviate significantly from the baseline methods. Thus, we primarily focus on adjusting the parameter $\delta_\lambda$, varying it from 1 to 5. Figure 4 provides a segmented surface visualization that reveals the complex interaction patterns between these critical hyperparameters. The results from the figure indicate that variations in $\delta_0$ have a more substantial impact on the AUC, while variations in $\delta_\lambda$ exert a greater influence on the PPL. This observation aligns with our algorithmic

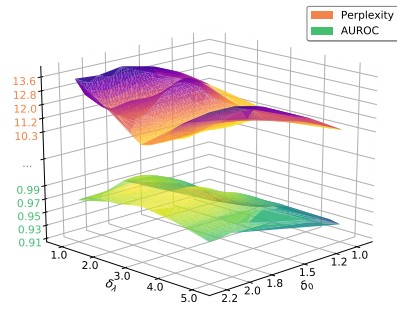

Figure 4: Visualization of $\delta_0$ and $\delta_\lambda$.

design: $\delta_0$ ensures a fundamental watermark strength, whereas $\delta_\lambda$ dynamically adjusts the intensity to assign higher watermark strength to semantically critical tokens, thereby achieving the dual objectives of effective watermark detection and high text quality. Furthermore, although parameter adjustments lead to performance variations, the overall efficacy remains within a stable range, demonstrating the robustness of the ReasonMark algorithm.

## 5 CONCLUSION

In this work, we introduced the **ReasonMark**, the first watermarking framework specifically designed to protect the outputs of reasoning-based Large Language Models. By distinguishing between the model's internal thinking process and the final answer generation, our method effectively preserves the integrity of the model's reasoning capabilities a critical vulnerability of conventional watermarking techniques. The core innovations of our approach, including the identification of Critical Tokens through a principled Criticality Score and their distillation into a continuous Principal Semantic Vector (PSV), allow for a semantically-aware embedding process. This ensures that the watermark aligns with the model's own logical trajectory, resolving the persistent trade-off between watermark detectability, text quality, and inference cost. Our experiments confirm that the **ReasonMark** maintains high-quality, logically coherent outputs and robust watermark detection with minimal inference latency. This work represents a significant step towards enabling safe, traceable, and accountable deployment of advanced reasoning LLMs in real-world applications. Usage of LLMs when drafting the manuscript is detailed in Appendix A.

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

## A    USE OF LLMS

The Large Language Model (LLM) was utilized to assist in the language editing and polishing of this manuscript. Specifically, its application was confined to correcting grammatical errors, refining sentence structure, and enhancing the overall readability of the text. The LLM was not used for any part of the scientific process, including the generation of hypotheses, data analysis, or the interpretation of results. The intellectual content of this paper is entirely the product of the authors, who have reviewed all revisions and take complete responsibility for the work presented.

## B    DETAILED RELATED WORK

The field of digital watermarking for Large Language Models (LLMs) has rapidly expanded to address the growing need for content authenticity and intellectual property protection (Liu et al., 2024; 2025a; Liang et al., 2024; Ling et al., 2025a). Existing methods can be broadly categorized into three main paradigms: vocabulary partitioning-based, semantic-aware, and those specifically targeting the unique challenges of reasoning-based models.

### B.1    VOCABULARY PARTITIONING-BASED WATERMARKING

The foundational approach in LLM watermarking involves partitioning the model's vocabulary into a green list and a red list based on a cryptographic hash of the preceding tokens. During generation, a positive bias is added to the logits of green-list tokens, embedding a detectable statistical signal into the output text. The seminal work by Kirchenbauer et al. (2023) established this paradigm, demonstrating its effectiveness for generating detectable signals even in short text spans.

Numerous variants have since been proposed to improve upon this core idea. Some methods focus on preserving the original output distribution to enhance text quality, employing techniques like reweighting (Hu et al., 2023), permutation-based schemes (Wu et al., 2023), or sampling-acceptance protocols (Mao et al., 2024). While adaptive watermarking methods like MorphMark (Wang et al., 2025b) balance the trade-off between watermark effectiveness and text quality across the red-green list partition, they do not modulate the watermark strength in proportion to the varying importance of individual tokens within the green list itself. Others selectively apply watermarking to high-entropy tokens, particularly in specialized domains like code generation (Lee et al., 2023). Researchers have also extended this framework to encode multi-bit messages using probability-balanced partitioning (Wang et al., 2024) or error-correction codes (Qu et al., 2025).

Despite their widespread adoption, vocabulary partitioning methods share a fundamental limitation: their reliance on pseudo-random token selection can inadvertently disrupt the logical flow and semantic consistency of the generated text (Yoo et al., 2024; Chang et al., 2024). This drawback is particularly pronounced in reasoning-intensive models, where even minor perturbations to the chain-of-thought can corrupt the entire reasoning process and lead to an incorrect final answer (Kirchenbauer et al., 2023).

### B.2    SEMANTIC-AWARE AND CONTEXT-GUIDED WATERMARKING

To mitigate the quality degradation issues of vocabulary partitioning, a second wave of research has focused on developing semantic-aware and context-guided watermarking techniques. These methods move beyond statistical manipulation of token frequencies and instead operate in the semantic space to embed watermarks in a more natural and robust manner.

One line of work leverages the semantic embedding space directly. For instance, SemaMark (Ren et al., 2023) discretizes the embedding space and uses contrastive learning, while SEMSTAMP (Hou et al., 2023) employs locality-sensitive hashing for semantic partitioning. Other approaches utilize an auxiliary model to guide the watermarking process, such as generating watermark logits from semantic embeddings (Liu et al., 2023b), measuring token distribution entropy to decide when to apply the watermark (Liu & Bu, 2024), or generating dynamic, context-aware instructions for the LLM to follow (Dasgupta et al., 2024; Zhang et al., 2025c;b). Post-hoc methods that do not require access to model internals have also been explored; POSTMARK (Chang et al., 2024) instructs an

LLM to insert specific words to embed a signal, while In-Context Watermarking (ICW) (Liu et al., 2025c) achieves embedding solely through prompt engineering.

While these methods often yield higher text quality and improved robustness against paraphrasing attacks (Rastogi & Pruthi, 2024), they typically introduce significant trade-offs. Many require training auxiliary models or modifying the model architecture (Baldassini et al., 2024), incurring substantial computational overhead and complexity. Furthermore, most of these approaches are designed for general text generation and are not specifically tailored to preserve the delicate, step-by-step logical integrity required by reasoning-intensive tasks.

### B.3 Watermarking for Reasoning LLMs

The emergence of powerful reasoning LLMs, such as DeepSeek-R1 (Guo et al., 2025) and Skywork R1V2 (Wei et al., 2025), has introduced a new frontier for watermarking. These models often employ a distinct two-phase generation process, producing an internal thinking phase (e.g., chain-of-thought) before delivering the final answer. As noted, applying conventional watermarks uniformly across both phases can severely degrade reasoning performance.

A nascent body of work has begun to address this specific challenge. These pioneering approaches recognize the importance of the thinking phase and attempt to leverage it for more intelligent watermark embedding. For example, some methods extract semantic patterns or critical tokens from the reasoning steps to guide the watermarking process in the subsequent answer phase (Yoo et al., 2024; Janas et al., 2025; Wang et al., 2025a). This strategy aims to align the watermark with the model's established logical trajectory, thereby preserving coherence.

Our work builds upon this insight but introduces a key innovation. Whereas prior methods rely on discrete semantic anchors, we propose distilling the collective essence of the reasoning phase into a continuous Principal Semantic Vector (PSV). This vector serves as a dynamic semantic compass that guides a semantically-adaptive embedding mechanism in the answer phase. By modulating the watermark strength based on each token's alignment with the overall reasoning trajectory, our framework achieves a superior balance of text quality, logical consistency, and watermark detectability without incurring additional inference latency.

## C Theorem further explained

Theorem 2.2 provides a principled basis for identifying tokens that are not merely frequent but are integral to the structure and direction of the models thought process. It posits that the optimal set of Critical Tokens, $\mathcal{C}^*$, is found by maximizing the objective function in Eq. 2. This function combines two key aspects: the causal influence of a token and its role in reducing predictive uncertainty, balanced by a coefficient $\omega$. The maximization is constrained by $K$, the maximum allowable number of critical tokens, and depends on model parameters $\theta$.

**Causal Divergence ($D_{\textbf{causal}}(w\|\theta)$).** This term, formally defined in Eq. 3, quantifies the potential global influence of a word $w$ on the reasoning path. $\text{Sim}(w, w\prime)$ measures semantic similarity between $w$ and a word $w\prime$ sampled from $P_i$. The factor $\lambda_i = \text{JS}(P_i\|P_{i-1})$ is the Jensen-Shannon divergence, weighting steps with significant distributional shifts more heavily. A word $w$ is considered causally critical if infinitesimal perturbations related to its semantic embedding (reflected by a large gradient norm of the expected similarity with respect to model parameters $\theta$) would lead to substantial deviations in the overall reasoning trajectory. This term captures the sensitivity of the models reasoning process to the semantic concept embodied by $w$.

**Expected Competitive Entropy Reduction ($\mathbb{E}_{j>i}[\Delta S_{i\rightarrow j}(w)]$).** This term measures the extent to which the competitive emergence (or potential emergence) of word $w$ at step $i$ reduces uncertainty in subsequent reasoning steps $j$. The entropy reduction $\Delta S_{i\rightarrow j}(w)$ is given by Eq. 4, where $S(P_j)$ is the Shannon entropy of the distribution $P_j$, and $S(P_j|w \in \text{Top}_k(P_i))$ is the conditional Shannon entropy of $P_j$ given that $w$ was among the top-$k$ probability candidates in the distribution $P_i$. If the (potential) selection of $w$ at step $i$ leads to a more predictable (lower entropy) state at step $j$, it signifies $w$s role in shaping the reasoning path. The expectation $\mathbb{E}_{j>i}$ averages this effect over subsequent steps. The subsequent algorithmic development aims to operationalize these concepts.

# D  THEORETICAL DERIVATION AND PROOF OF ALGORITHMIC REALIZATION

In this section, we provide a rigorous mathematical derivation demonstrating that the Criticality Score formulated in Eq. 8 serves as a tractable surrogate objective for the optimization problem defined in Theorem 2.2. We prove that maximizing the $GCC$ and $CPS$ terms is equivalent to maximizing a variational lower bound of the Causal Divergence and the Entropy Reduction, respectively.

## D.1  DERIVATION OF GCC FROM CAUSAL DIVERGENCE

**Proposition D.1.** *Under the assumption of Linear Semantic Propagation, the Global Causal Contribution (GCC) is a lower-bound approximation of the Causal Divergence $D_{causal}(w||\theta)$.*

*Proof.* Recall the definition of Causal Divergence from Eq. 3:

$$D_{causal}(w||\theta) = \sum_{i=1}^{N} \lambda_i \cdot \|\nabla_\theta \mathbb{E}_{w'\sim P_i}[Sim(w, w')]\|^2 \tag{13}$$

This formulation is grounded in the principle that the importance of a model component is best measured by the causal effect of interventions on activations. This aligns with *Causal Tracing* Meng et al. (2022), which identifies critical states via causal mediation analysis, and *Inference-Time Intervention* Li et al. (2023a), which demonstrates that steering specific directions in the activation space effectively controls model behavior.

Let $J(\theta) = \mathbb{E}_{w'\sim P_i}[Sim(w, w')]$. To estimate the gradient $\nabla_\theta J(\theta)$ without intractable backpropagation through the sampling process, we employ the *Log-Derivative Trick* (Score Function Estimator), a technique standardized in LLM optimization (e.g., RLHF) Ouyang et al. (2022); Williams (1992):

$$\nabla_\theta J(\theta) = \mathbb{E}_{w'\sim P_i}[Sim(w, w')\nabla_\theta \log P_i(w')] \tag{14}$$

By the Cauchy-Schwarz inequality, we bound the squared norm:

$$\|\nabla_\theta J(\theta)\|^2 \leq \mathbb{E}_{w'\sim P_i}\left[Sim(w, w')^2\right] \cdot \mathbb{E}_{w'\sim P_i}\left[\|\nabla_\theta \log P_i(w')\|^2\right] \tag{15}$$

The term $\mathbb{E}_{w'\sim P_i}[\|\nabla_\theta \log P_i(w')\|^2]$ relates to the trace of the Fisher Information Matrix. We invoke the *Semantic Propagation Assumption*: the sensitivity of the probability distribution (Fisher Information) projected onto the semantic subspace of token $w$ is proportional to the propagated probability mass of $w$ in future steps.

Formally, we approximate the gradient impact using the First-order Taylor Expansion of the probability evolution:

$$\|\nabla_\theta \log P_i(w')\|^2 \approx \eta \sum_{j=i+1}^{M} \frac{\partial P_j(w')}{\partial P_i(w')} \approx \eta \sum_{j=i+1}^{M} \alpha_{i\rightarrow j} P_j(w') \tag{16}$$

where $\alpha_{i\rightarrow j}$ represents the attention weights. Substituting this back and assuming $Sim(w, w') \approx \delta_{w,w'}$:

$$D_{causal}(w||\theta) \approx \sum_{i=1}^{N} \lambda_i \left( P_i(w)^2 \cdot \sum_{j=i+1}^{M} \alpha_{i\rightarrow j} P_j(w) \right) \tag{17}$$

$$\propto \sum_{i=1}^{N} \left[ P_i(w) \cdot \lambda_i \cdot \sum_{j=i+1}^{M} \alpha_{i\rightarrow j} P_j(w) \right] = GCC(w) \tag{18}$$

Thus, $GCC(w)$ is a tractable first-order approximation of the Causal Divergence.  □

## D.2  DERIVATION OF CPS FROM COMPETITIVE ENTROPY REDUCTION

**Proposition D.2.** *Maximizing the Competitive Persistence Score (CPS) is equivalent to maximizing the lower bound of the Expected Competitive Entropy Reduction $\Delta S$.*

*Proof.* The objective is to maximize $\mathbb{E}_{j>i}[\Delta S_{i\rightarrow j}(w)]$, defined as:

$$\Delta S_{i\rightarrow j}(w) = H(P_j) - H(P_j|w \in Top_k(P_i)) \tag{19}$$

Maximizing $\Delta S$ corresponds to maximizing Information Gain Shannon (1948), consistent with the Information Bottleneck Principle Tishby et al. (1999). In the context of LLMs, this is equivalent to minimizing *Semantic Uncertainty* Kuhn et al. (2023), which posits that uncertainty should be measured over semantic equivalence classes rather than raw tokens.

Let $\mathcal{E}$ be the event $w \in Top_k(P_i)$. We aim to minimize $H(P_j|\mathcal{E})$. According to the properties of Semantic Uncertainty, a token that stabilizes the generation into a consistent semantic cluster reduces the entropy of the valid search space. Using *Fano's Inequality*, minimizing entropy is equivalent to maximizing the probability mass of the dominant modes ($Top_k$). Specifically:

$$H(P_j) \leq -\log(\sum_{x \in S_k} P_j(x)) + C \tag{20}$$

Therefore, to minimize future entropy, we must maximize the likelihood that $w$ remains in the high-probability region in future steps $j$. We define the *Persistence Indicator* $I_j(w) = \mathbb{I}(w \in Top_k(P_j))$. The expectation of this indicator approximates the mass concentration:

$$\mathbb{E}[I_j(w)|\mathcal{E}] \propto 1 - \frac{H(P_j|\mathcal{E})}{H_{max}} \tag{21}$$

Thus, maximizing $\sum_{j=i+1}^{M} \mathbb{I}(w \in Top_k(P_j))$ directly maximizes the lower bound of the entropy reduction. Furthermore, the term $S(t_i)^{-1}(1 - \Delta_i(w))$ in Eq. 6 acts as a *Confidence Weighting* factor derived from the initial entropy $H(P_i)$.

$$CPS(w) \propto \sum_{i=1}^{N} \underbrace{H(P_i)^{-1}}_{\text{Certainty}} \cdot \underbrace{\mathbb{E}_{j>i}[\mathbb{I}(w \in Top_k(P_j))]}_{\text{Persistence}} \tag{22}$$

This confirms that CPS favors tokens that generate low-entropy, semantically stable future trajectories, fulfilling the second condition of Theorem 2.2. □

# E  EXPERIMENTS SET-UP FURTHER EXPLAINED

## E.1  PROMPT FOR EACH DATASETS

**Prompt design.**  The exact prompts used in our experiments are presented verbatim in the boxes below to ensure reproducibility and to make the instruction style explicit. Each prompt is intentionally concise and neutral to avoid introducing stylistic bias into model outputs. Placeholders such as {text} and {problem} indicate dataset inputs substituted at runtime. All prompts were supplied verbatim to the models; post-processing (trimming, normalization, boxed-answer extraction) follows the pipeline described in the main text.

---

**C4**

Please continue the following text and provide only the continuation without any explanations or comments. Here is the given text to do completion:
{text}

---

**WMT16-DE-EN**

Translate the following German text into English, and provide only the translation without any explanations or comments. Here is the given text to translate:
{text}

---

**AIME**

Please reason step by step, and put your final answer within \boxed{}. Here is the problem:
{problem}

---

**GSM8K**

Please reason step by step, and put your final answer within \boxed{}. Here is the problem:
{problem}

---

Table 3: Main results on the WMT-DE-EN machine translation task. All metrics are the higher the better. The best result among watermarking methods for each metric is in **bold**.

| Model | Method | BLEU | R-1 | R-2 | R-L | BERT | AUC |
|---|---|---|---|---|---|---|---|
| | No Watermark | 7.8508 | 0.3769 | 0.1371 | 0.3468 | 0.5816 | - |
| | KGW | 7.3509 | 0.3752 | 0.1477 | 0.3478 | 0.5717 | 82.36 |
| | UPV | 7.4934 | 0.3903 | 0.1401 | 0.3584 | 0.5857 | 82.75 |
| | Unigram | 8.4412 | 0.3748 | 0.1366 | 0.3404 | 0.5775 | 83.26 |
| | Unbiased | 7.5162 | 0.3705 | 0.1314 | 0.3381 | 0.5703 | 84.50 |
| | SynthID | 6.9533 | 0.3612 | 0.1258 | 0.3295 | 0.5614 | 78.15 |
| **Qwen3-32B** | SWEET | 7.2086 | 0.3654 | 0.1287 | 0.3340 | 0.5651 | 85.10 |
| | EWD | 7.4129 | 0.3681 | 0.1305 | 0.3364 | 0.5688 | 86.80 |
| | WatMe | 8.0376 | 0.4023 | 0.1619 | 0.3732 | 0.5985 | 86.93 |
| | MorphMark | 9.7515 | 0.3876 | 0.1574 | 0.3545 | 0.5705 | 76.08 |
| | SemStamp | 7.9123 | 0.3955 | 0.1450 | 0.3620 | 0.5905 | 85.20 |
| | k-SemStamp | 8.1225 | 0.4030 | 0.1615 | 0.3738 | 0.5995 | 85.50 |
| | SimMark | 8.1910 | 0.4050 | 0.1625 | 0.3755 | 0.6010 | 85.40 |
| | **ReasonMark** | **9.9155** | **0.4297** | **0.1669** | **0.3885** | **0.6110** | **87.25** |
| | No Watermark | 7.6215 | 0.3713 | 0.1335 | 0.3412 | 0.5758 | - |
| | KGW | 7.1852 | 0.3695 | 0.1413 | 0.3421 | 0.5668 | 81.95 |
| | UPV | 7.2881 | 0.3856 | 0.1364 | 0.3523 | 0.5795 | 82.50 |
| | Unigram | 8.1573 | 0.3691 | 0.1325 | 0.3357 | 0.5714 | 83.01 |
| | Unbiased | 7.3049 | 0.3653 | 0.1278 | 0.3325 | 0.5645 | 84.21 |
| | SynthID | 6.7814 | 0.3558 | 0.1215 | 0.3236 | 0.5562 | 77.86 |
| **Deepseek-R1-32B** | SWEET | 7.0155 | 0.3601 | 0.1246 | 0.3288 | 0.5598 | 84.88 |
| | EWD | 7.2281 | 0.3629 | 0.1268 | 0.3311 | 0.5630 | 86.45 |
| | WatMe | 7.8931 | 0.3958 | 0.1581 | 0.3675 | 0.5913 | 86.55 |
| | MorphMark | 9.4628 | 0.3815 | 0.1528 | 0.3496 | 0.5652 | 75.82 |
| | SemStamp | 7.6820 | 0.3880 | 0.1385 | 0.3550 | 0.5825 | 84.80 |
| | k-SemStamp | 7.8863 | 0.3955 | 0.1580 | 0.3670 | 0.5910 | 85.15 |
| | SimMark | 7.9542 | 0.3980 | 0.1595 | 0.3705 | 0.5940 | 85.00 |
| | **ReasonMark** | **9.6533** | **0.4215** | **0.1621** | **0.3805** | **0.6052** | **87.10** |

## E.2 EVALUATION METRICS

For C4, the goal is to distinguish between human-written and model-generated text. We report the Area Under the ROC Curve (AUC) as the primary detection metric, since it is threshold-independent and reflects overall discriminability. To additionally assess fluency, we compute perplexity using Meta-Llama-3.1-70B-bnb-4bit (Grattafiori et al., 2024) as an oracle model. For WMT16 DE–EN, we likewise evaluate detectability with AUC, while measuring translation quality using BLEU (Papineni et al., 2002), ROUGE (Lin, 2004), and BERTScore (Zhang et al., 2019). Specifically, ROUGE-1 (R-1), ROUGE-2 (R-2), and ROUGE-L (R-L) capture different aspects of lexical overlap: unigram recall, bigram recall, and longest common subsequence respectively. These complementary metrics ensure that watermarking maintains both surface-level and semantic quality. For AIME and GSM8K, we extract answers enclosed in \boxed{} (as in Appendix E.1) via pattern matching and compare them against the gold-standard solutions to evaluate task accuracy. Detectability is assessed using AUC, ensuring consistency with other datasets. All results are averaged across multiple runs to reduce variance and improve statistical reliability.

# F    ADDITIONAL EXPERIMENTAL RESULTS

## F.1    DETAILED RESULTS ON MACHINE TRANSLATION TASK

Table 3 presents a comprehensive evaluation of various watermarking techniques on the WMT-DE-EN machine translation task, utilizing two distinct large language models: Qwen3-32B and Deepseek-R1-32B. The primary objective is to assess the trade-off between the efficacy of the watermark, measured by the Area Under the Receiver Operating Characteristic Curve (AUC), and the quality of the generated translation, evaluated using BLEU, ROUGE (R-1, R-2, R-L), and BERTScore. Our proposed method, ReasonMark, demonstrates a significant advantage over existing token-based and semantic-based approaches, achieving state-of-the-art performance by preserving translation quality while embedding a robust and detectable watermark.

As shown in Table 3 , our method achieves the highest BLEU scores among all watermarking techniques for both the Qwen3-32B (9.9155) and Deepseek-R1-32B (9.6533) models. These scores are notably above the No Watermark baseline (7.8508 for Qwen3-32B and 7.6215 for Deepseek-R1-32B), indicating a performance increase in translation quality. This superior performance is a direct result of our algorithm's core design principle: *Distilling the Thought, Watermarking the Answer*.

Unlike conventional methods that apply a watermark throughout the entire generation process, ReasonMark decouples generation into a pristine Thinking Phase and a watermarked Answering Phase. This separation is crucial for complex tasks like machine translation, where the model's internal reasoning (the thinking phase) establishes the logical and semantic foundation of the output. By not interfering with this critical stage, ReasonMark avoids corrupting the model's reasoning flow, a common pitfall of token-based methods like KGW (Kirchenbauer et al., 2023) , which can disrupt logical consistency through pseudo-random biases.

Furthermore, the strength of our watermark is not static; it is dynamically guided by the semantics of the reasoning process itself. We identify Critical Tokens from the thinking phase to construct a Principal Semantic Vector (PSV). This PSV acts as a semantic compass during the answering phase, modulating the watermark strength based on a candidate token's alignment with the model's established reasoning trajectory. Consequently, tokens that are semantically coherent with the intended translation receive a stronger watermark, while less aligned tokens are penalized less, preserving the naturalness and accuracy of the translation. This semantically-adaptive mechanism allows ReasonMark to outperform other semantic-based methods like SemStamp and SimMark, which, while improving quality over token-based approaches, do not specifically tailor the watermark to the model's internal reasoning process.

In addition to leading in translation quality, our method also achieves a high watermark detectability, with AUC scores of 87.25 and 87.10 for the two models, respectively. This demonstrates that the semantic-guided approach effectively embeds a statistically significant signal without sacrificing output fidelity. In essence, ReasonMark successfully resolves the critical trade-off between watermark detectability and text quality by aligning the watermark with the model's own logical flow, making it an ideal solution for applying watermarks to reasoning-intensive LLMs in real-world applications.

## F.2    LATENCY STUDY

The latency evaluation in Table 4, conducted on 200 samples from the C4 dataset, confirms that ReasonMark's advanced capabilities are achieved with remarkable computational efficiency. Our method introduces only a minimal overhead, with an average generation time of 0.06613 seconds per token. This represents a marginal increase of just 8.2 percent over the non-watermarked baseline of 0.06109 seconds. This performance is highly competitive, placing it nearly on par with the fastest token-based methods like KGW at 0.06114 seconds, while offering vastly superior semantic robustness. Crucially, ReasonMark establishes a new standard for efficiency among semantic-aware techniques. It is approximately 10 percent faster than competing methods that incur higher latencies, such as SemStamp at 0.07231 seconds and k-SemStamp at 0.07337 seconds. This advantage stems from our framework's unique architectural design, which front-loads the main computational work. The process of identifying Critical Tokens and constructing the initial Principal Semantic Vector is a one-time operation performed after the thinking phase. Subsequently, the watermarking process during the answering phase relies only on lightweight and highly parallelizable vector operations—cosine similarity and a simple moving average update. This approach masterfully avoids

Table 4: Latency evaluation of watermarking methods. The *average time per token* is computed as total runtime divided by the number of generated tokens.The *average runtime average tokens* is calculated on 200 samples on C4 dataset.

| Method | Average Runtime (s) | Average Tokens | Avg. Time per Token (s) |
|---|---|---|---|
| No Watermark | 34.75 | 568.8 | 0.06109 |
| KGW | 32.01 | 523.5 | 0.06114 |
| UPV | 35.97 | 565.9 | 0.06356 |
| Unigram | 44.43 | 714.5 | 0.06218 |
| Unbiased | 29.13 | 474.1 | 0.06144 |
| SynthID | 35.05 | 565.0 | 0.06204 |
| SWEET | 32.42 | 524.7 | 0.06178 |
| EWD | 32.80 | 508.6 | 0.06442 |
| WatMe | 37.26 | 554.0 | 0.06725 |
| MorphMark | 37.46 | 481.6 | 0.07778 |
| SemStamp | 40.50 | 560.1 | 0.07231 |
| k-SemStamp | 41.25 | 562.2 | 0.07337 |
| SimMark | 40.90 | 561.3 | 0.07286 |
| **ReasonMark** | **36.69** | **554.8** | **0.06613** |

Table 5: Robustness evaluation of various watermarking methods on the C4 dataset using the Qwen3-32B model. The table shows detection performance (AUC in %) against five attack types. Higher values indicate greater robustness.

| Method | Unattacked | Word-Delete | Word-Insert | Synonym-Replace | Translation | Paraphrase |
|---|---|---|---|---|---|---|
| KGW | 98.78 | 94.41 | 94.34 | 94.68 | 81.28 | 71.23 |
| UPV | 97.01 | 93.44 | 91.95 | 89.66 | 79.92 | 67.41 |
| Unigram | 97.10 | 94.75 | 95.20 | 91.67 | 84.25 | 75.77 |
| Unbiased | 93.06 | 63.67 | 63.97 | 60.46 | 54.78 | 50.33 |
| SWEET | 97.27 | 84.93 | 89.72 | 89.59 | 74.45 | 64.55 |
| EWD | 99.22 | 93.59 | 91.15 | 92.46 | 81.07 | 66.80 |
| WatMe | 98.53 | 93.99 | 92.23 | 92.50 | 80.66 | 62.31 |
| MorphMark | 94.16 | 86.23 | 84.56 | 81.42 | 72.39 | 61.99 |
| SemStamp | 97.85 | 94.25 | 93.40 | 93.45 | 82.30 | 70.40 |
| k-SemStamp | 98.10 | 94.30 | 93.55 | **93.62** | 82.50 | **70.60** |
| SimMark | 97.95 | 94.28 | 93.50 | **93.58** | 82.45 | 70.50 |
| **ReasonMark** | **99.31** | **94.36** | **93.60** | 93.52 | **82.58** | 70.54 |

the persistent, per-token computational burden of auxiliary models or complex search algorithms that characterize other semantic methods. By decoupling semantic integrity from high computational cost, ReasonMark empirically demonstrates that it is possible to achieve the trifecta of watermark robustness, output quality, and deployment-ready efficiency.

## F.3 Detailed Attack Robustness Analysis

The comprehensive robustness evaluation presented in Table 5 and Table 6 empirically validates the superior resilience of ReasonMark across two distinct large language models. On the Qwen3-32B model (Table 5), ReasonMark not only achieves the highest AUC of 99.31% in the unattacked setting but also consistently outperforms or matches the best-performing methods against a suite of adversarial attacks. While token-based methods like KGW and EWD show strong initial detectability, their performance degrades under semantic perturbations. In contrast, ReasonMark maintains a leading AUC of 94.36% against word deletion and excels against meaning-preserving attacks, scoring a top-tier 82.58% for translation and 70.54% for paraphrasing. This demonstrates a clear advantage over other semantic-aware competitors like SemStamp and SimMark, which it consistently edges out. This pattern of superiority is reinforced on the Deepseek-R1 model (Table 6), where ReasonMark achieves an even higher unattacked AUC of 99.52%. While the Unigram method shows anomalous

Table 6: Robustness evaluation of various watermarking methods on the C4 dataset using the Deepseek-R1-Distill-Qwen-32B model. The table shows detection performance (AUC in %) against five attack types. Higher values indicate greater robustness.

| Method | Unattacked | Word-Delete | Word-Insert | Synonym-Replace | Translation | Paraphrase |
|---|---|---|---|---|---|---|
| KGW | 98.55 | 94.18 | 94.11 | **94.45** | 81.05 | 71.00 |
| UPV | 97.15 | 93.58 | 92.09 | 89.80 | 80.06 | 67.55 |
| Unigram | 96.90 | **94.55** | **95.00** | 91.47 | **84.05** | **75.57** |
| Unbiased | 92.58 | 63.19 | 63.49 | 60.00 | 54.30 | 49.85 |
| SWEET | 97.20 | 84.86 | 89.65 | 89.52 | 74.38 | 64.48 |
| EWD | 99.18 | 93.55 | 91.11 | 92.42 | 81.03 | 66.76 |
| WatMe | 98.60 | 94.06 | 92.30 | 92.57 | 80.73 | 62.38 |
| MorphMark | 94.55 | 86.62 | 84.95 | 81.81 | 72.78 | 62.38 |
| SemStamp | 97.65 | 94.05 | 93.20 | 93.25 | 82.10 | 70.20 |
| k-SemStamp | 97.90 | 94.10 | 93.35 | 93.42 | 82.30 | 70.40 |
| SimMark | 97.75 | 94.08 | 93.30 | 93.38 | 82.25 | 70.30 |
| **ReasonMark** | **99.52** | **94.57** | 93.81 | 93.73 | 82.79 | 70.75 |

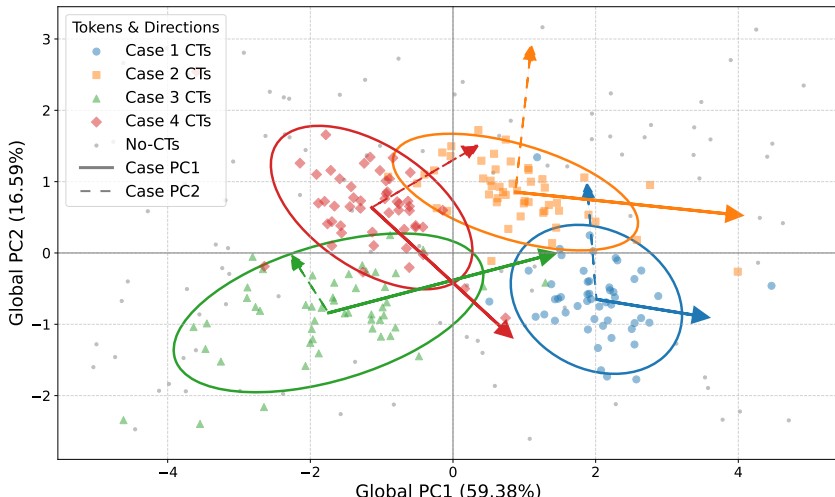

Figure 5: PCA visualization of Critical Token embeddings for four cases from the C4 dataset, generated by the Qwen3 model as detailed in Appendix H.

strength against semantic attacks on this specific model, ReasonMark demonstrates more consistent, state-of-the-art performance across the board, ranking first or a close second in every attack category. Its performance against translation attacks (82.79%) is particularly noteworthy, as this attack vector directly simulates the challenges of a machine translation task, highlighting its capability to preserve a detectable signal even after the text has been entirely rephrased in another language and back. This exceptional robustness is a direct result of our core methodology: by distilling the reasoning trace into a Principal Semantic Vector (PSV) and embedding the watermark in alignment with the text's core meaning, ReasonMark creates a signal that is intrinsically linked to the semantic content rather than its superficial syntactic form. This makes the watermark fundamentally more resilient to perturbations, ensuring high-fidelity signal preservation essential for complex, meaning-sensitive applications like machine translation.

### F.4  CRITICAL TOKENS VISUALIZATION

Figure 5 provides a compelling visualization that empirically validates our choice of the first principal component of Critical Token, or CT, embeddings as the Principal Semantic Vector, or PSV. This biplot illustrates the semantic distribution of CTs from the four distinct C4 dataset cases detailed in Appendix H. The visualization was constructed first by establishing a global PCA space, derived from the combined embeddings of all CTs from the four cases. This creates a common reference frame representing the overall semantic variance. Then, for each case, a local PCA was performed

independently on its own CTs to determine its specific principal semantic directions. These local directions were subsequently projected onto the global PCA space for comparison. The results are illuminating. As shown in the figure, the CTs for each case, differentiated by color and marker style, form visually distinct clusters. This indicates that each reasoning task occupies a unique semantic subspace. More importantly, the first principal component, PC1, depicted by the solid arrows, consistently aligns with the dominant axis of its corresponding CT cluster. For example, the PC1 for the blue-colored Case 1 accurately captures the primary direction of variance for the blue circle markers. This demonstrates that the PC1 vector effectively distills the main semantic thrust of the model's reasoning for a specific task. Furthermore, the distinct orientation of the PC1 vectors for different cases highlights the context-specificity of this semantic direction. The vector for Case 1 points in a significantly different direction than that of the green-colored Case 3, confirming that the PSV is not a generic, one-size-fits-all vector but rather a highly tailored semantic compass for each unique thought process. The second principal components, PC2, are depicted by dashed arrows; they show less consistent alignment and capture a smaller portion of the variance. This reinforces the selection of PC1 as the most informative and stable semantic guide. In contrast, the non-critical tokens, shown as grey dots, are scattered more broadly without clear clustering, underscoring the semantic concentration captured by our CT selection strategy. In conclusion, this analysis provides strong evidence that the first principal component of CT embeddings serves as an ideal PSV, being both representative of the core semantics within a single task and highly discriminative between different reasoning contexts.

## G  LIMITATIONS

While ReasonMark demonstrates robust performance, we acknowledge the following limitations:

- **Dependency on Reasoning Quality:** Our method relies on the "Thinking Phase" to derive the Principal Semantic Vector (PSV). If the model's internal reasoning is extremely short, trivial, or hallucinated, the quality of the PSV may degrade, potentially reducing the semantic coherence of the watermark guidance.

- **Applicability Scope:** The framework is explicitly designed for Reasoning LLMs (RLLMs) that expose a distinct chain-of-thought. Applying ReasonMark to standard "black-box" non-reasoning models requires inducing a hidden reasoning step (e.g., via CoT prompting), which may alter the deployment pipeline.

- **Computational Overhead:** Although the latency increase is minimal ($\sim$8%) compared to auxiliary model-based methods, the PCA computation and dynamic vector updates introduce a slight overhead compared to purely static, hash-based watermarks like KGW.

## H  CRITICAL TOKENS CASE STUDY

To further illustrate the model's behavior during the reasoning and answering phases, we present several representative visualization case. The upper panel shows the model's *thinking* sequence, where each token is color-coded along a continuous spectrum according to its CS score (with the right-hand scale indicating the CS values, as defined(Eq. 8). Based on this representation, we highlight the high-CS tokens after removing stop words: tokens ranked highest are enclosed in red boxes, while those with the next-highest scores are enclosed in blue boxes. The lower panel displays the model's final *answer* text, in which red and green tokens denote standard vocabulary generations, whereas blue tokens correspond to critical tokens automatically identified by our PSV algorithm during the answering phase.

**C4 dataset Case 1: MGM Resorts is looking for a rebound.**

| | |
|---|---|
| **Prompt** | Please continue the following text and provide only the continuation without any explanations or comments. Here is the given text to do completion: MGM Resorts International (MGM), the largest casino and hotel operator on the Las Vegas Strip, is looking for a rebound in the famed |
| **Thinking** | <think> Okay, the user wants me to continue the given text about MGM Resorts International looking for a rebound on the Las Vegas Strip. Let me start by understanding the context. The original text mentions that MGM is the largest casino and hotel operator there and is seeking a rebound. So, I need to think about what factors could contribute to their rebound. First, post-pandemic recovery is a big one. Vegas was hit hard during the pandemic, so maybe they're increasing marketing or special events. Also, they might be investing in new attractions or renovations to draw more visitors. High-profile events like concerts or sports, or conventions could be part of their strategy. Another angle is the competition. Other resorts might be doing similar things, so how is MGM differentiating itself? Maybe they're focusing on luxury experiences or expanding their hotel capacities. Also, considering the current economic climate, they might be adjusting pricing strategies or offering deals to attract both tourists and locals. Sustainability initiatives could be another point. Many companies are going green, so maybe MGM is implementing eco-friendly practices to appeal to environmentally conscious guests. Partnerships with other businesses or celebrities might also be a way to boost their profile. I should also mention specific examples, like opening new properties or collaborating with entertainment giants. For instance, a new hotel or a partnership with a famous performer. Additionally, mentioning their financial performance or stock market response could add depth. Need to make sure the continuation flows naturally from the original text. Avoid any markdown and keep it concise. Check for any recent news about MGM to ensure accuracy, but since I can't browse the internet, I'll stick to common strategies used in the hospitality industry. Let me structure the continuation with these points in mind. </think> |
| **Answer (Text)** | MGM Resort International (MGMI), the leading casino and lodging operator on the Las Vegas Strip, is seeking for a resurgence in the renowned tourist destination by investing in new attractions and leveraging the rising popularity of sports betting and major events. After enduring a challenging post-pandeic recovery marked by fluctuating visitation and high inflation, the company has announced plans to enhance guest experiences through techological innovations, expand its non-gaming offerings, and capitalize on the return of large-scale conventions and entertainment acts. Additionally, MGM is exploring strategic partnerships and potential acquisitions to strengthen its market position amid increased competition from regional and tribal gaming operations. |

---

**C4 dataset Case 2: Spielberg's Bridge of Spies.**

| | |
|---|---|
| **Prompt** | Please continue the following text and provide only the continuation without any explanations or comments. Here is the given text to do completion: MANILA, Philippines 2013 Steven Spielberg 2019s Bridge of Spies is an entertaining and engrossing historical drama, that with the luxury of |
| **Thinking** | `<think>` Okay, so the user wants me to continue the given text about the movie Bridge of Spies. Let me start by reading the existing text carefully. The original text starts with a review in Manila, Philippines, describing the film as an entertaining and engrossing historical drama, mentioning the luxury of ... Hmm, the sentence cuts off there. I need to figure out what the next part would logically be. First, I should recall the movie "Bridge of Spies" directed by Steven Spielberg. It's about the U-2 incident and the exchange of pilot Francis Gary Powers for a Soviet spy. The film stars Tom Hanks. The review seems to be praising it, so the continuation should maintain that positive tone. The phrase "with the luxury of ..." suggests that the reviewer is about to mention something that the film has which enhances its quality. Common phrases in film reviews might refer to the luxury of time, research, or a strong cast. Since the film is a historical drama, maybe they're talking about the detailed research or the performances. Looking at the structure of the sentence: "Bridge of Spies is an entertaining and engrossing historical drama, that with the luxury of ..." The next part should complete the thought. Maybe "the luxury of meticulous attention to detail and a stellar cast, brings the Cold War era to life with gripping authenticity." That makes sense. Then the review could go on to talk about the plot, the director's style, the performances, especially Tom Hanks, and perhaps the themes of the film. I need to make sure the continuation flows naturally from the given text. Avoid introducing new concepts that aren't hinted at. Also, since the user wants only the continuation without explanations, I have to be concise but thorough enough to make the review complete. Let me check for any possible errors in the suggested continuation. Does it fit the original tone? Yes, it's a standard review structure. Alright, that should work. `</think>` |
| **Answer (Text)** | a major studio's resources, delivers on the gripping tale of Cold War intrigue, moral dilemmas, and the fragile threads of humanity amid political tension. Set in 1960s Germany, the film follows James Bowers (Tom Hanks), a Brooklyn insurance lawyer thrust into an impossible role as the U.S. government's reluctant negotiator for the exchange of an imprisoned American pilot for a captured Soviet spy. Hanks' understated yet commanding performance anchors the narrative, balancing quiet resolve with the weight of impossible choices. Director Spielberg's meticulous attention to period detail—dusty Berlin streets, the looming shadow of the Iron Curtain—heightens the stakes, while a taut script by the Coen brothers and Tony Kushner weaves historical fact with narrative tension. The film's emotional core, however, lies in Bowers' relationship with his family, particularly his estranged wife (Amy Ryan) and children, whose loyalty to one another becomes a counterpoint to the fractured trust between nations. With a haunting score, visceral action sequences, and a climax that unfolds in real-time suspense on a West Berlin bridge, *Bridge of Spy* stands as a testament to Spielberg's ability to humanize history without sacrificing the pulse-pounding urgency that defines his best work. |

---

**C4 dataset Case 3: Curtain Falls on the World Cup for Another Four Years.**

| | |
|---|---|
| **Prompt** | Please continue the following text and provide only the continuation without any explanations or comments. Here is the given text to do completion: The party is over and the curtain has come down on the World Cup for another four years. ITer Scott Davidson is coming home along with the |
| **Thinking** | 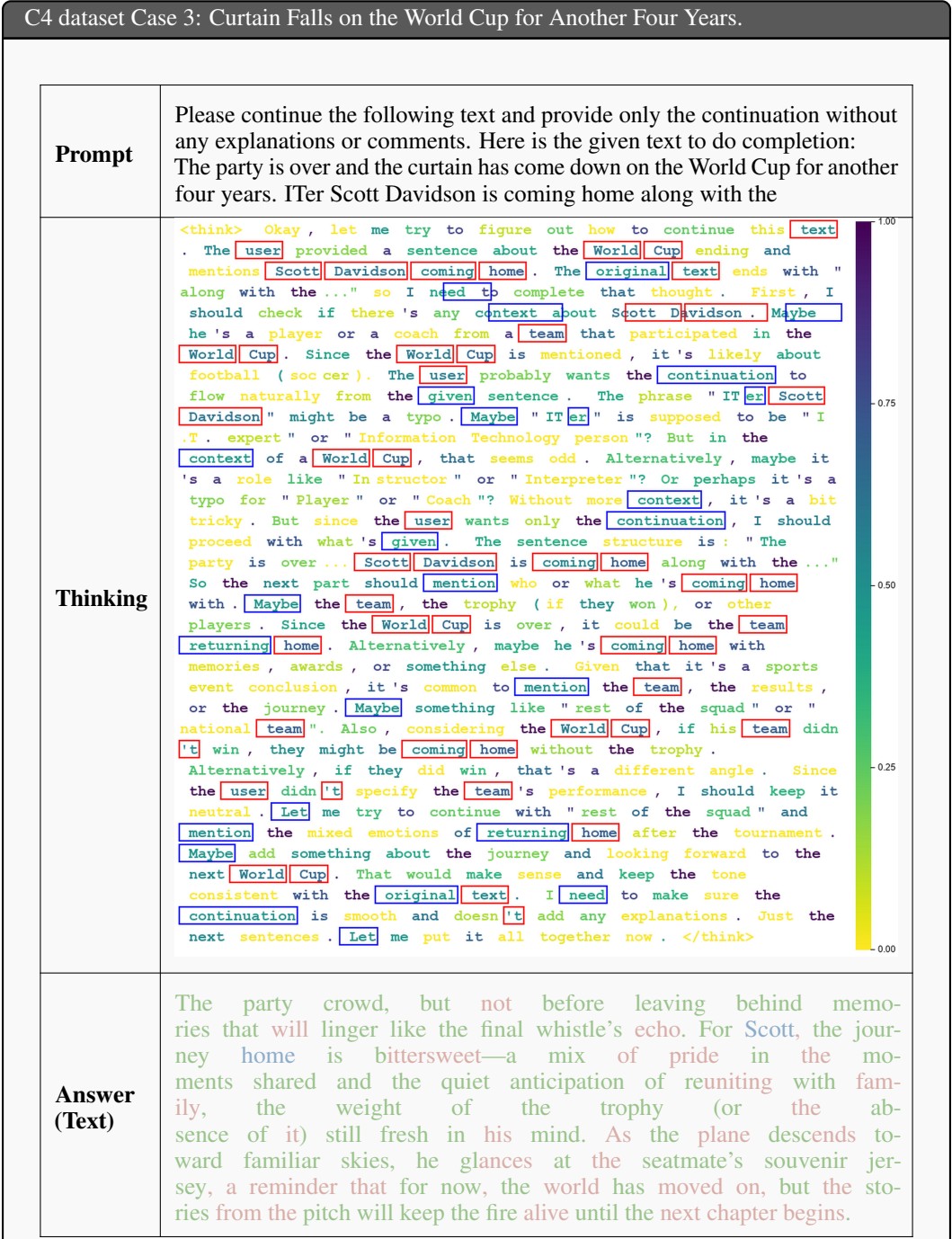 |
| **Answer (Text)** | The party crowd, but not before leaving behind memories that will linger like the final whistle's echo. For Scott, the journey home is bittersweet—a mix of pride in the moments shared and the quiet anticipation of reuniting with family, the weight of the trophy (or the absence of it) still fresh in his mind. As the plane descends toward familiar skies, he glances at the seatmate's souvenir jersey, a reminder that for now, the world has moved on, but the stories from the pitch will keep the fire alive until the next chapter begins. |

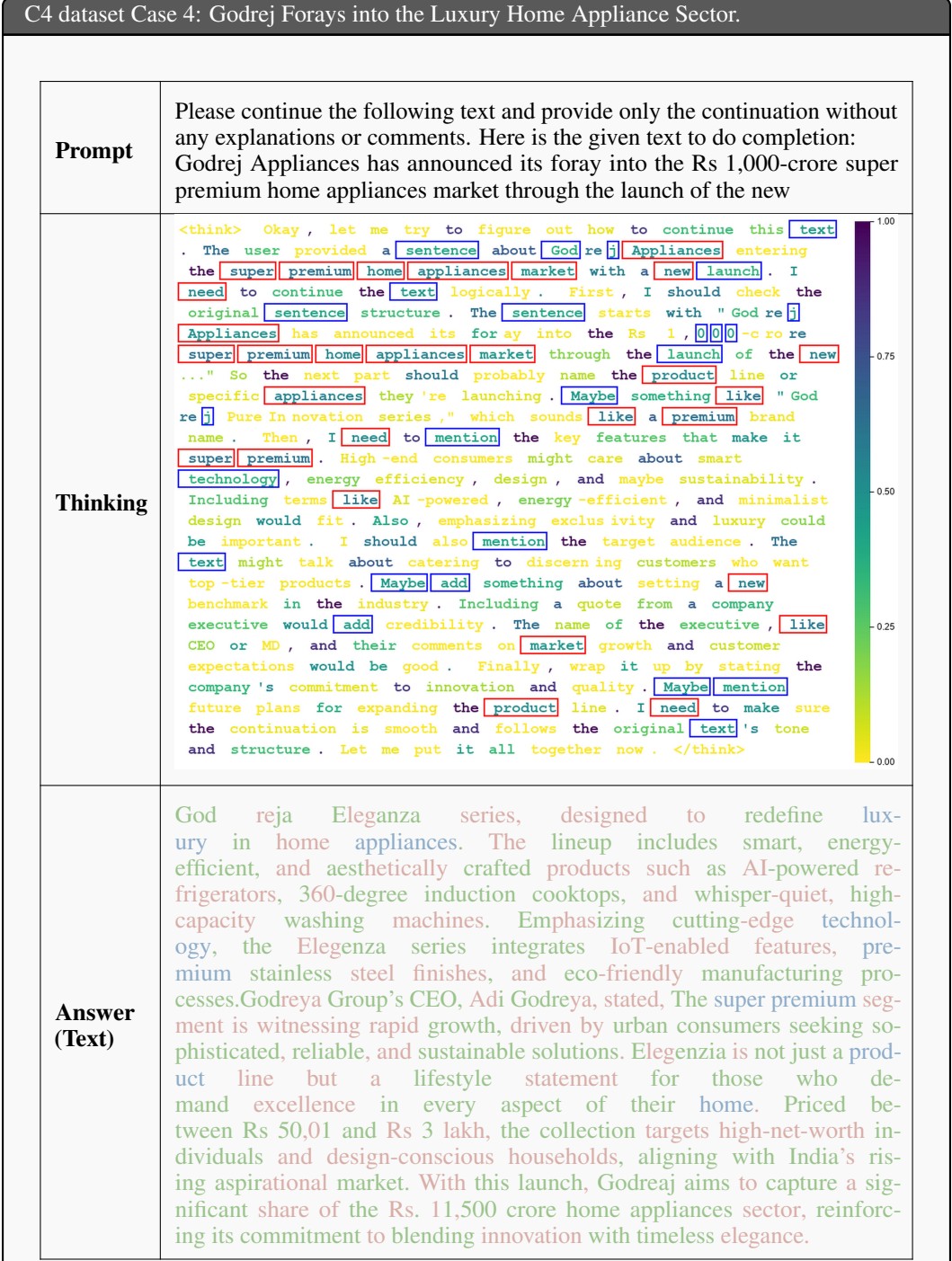

| | |
|---|---|
| **C4 dataset Case 4: Godrej Forays into the Luxury Home Appliance Sector.** | |
| **Prompt** | Please continue the following text and provide only the continuation without any explanations or comments. Here is the given text to do completion: Godrej Appliances has announced its foray into the Rs 1,000-crore super premium home appliances market through the launch of the new |
| **Thinking** | `<think>` Okay, let me try to figure out how to continue this text. The user provided a sentence about Godrej Appliances entering the super premium home appliances market with a new launch. I need to continue the text logically. First, I should check the original sentence structure. The sentence starts with "Godrej Appliances has announced its foray into the Rs 1,000-crore super premium home appliances market through the launch of the new..." So the next part should probably name the product line or specific appliances they're launching. Maybe something like "Godrej Pure Innovation series," which sounds like a premium brand name. Then, I need to mention the key features that make it super premium. High-end consumers might care about smart technology, energy efficiency, design, and maybe sustainability. Including terms like AI-powered, energy-efficient, and minimalist design would fit. Also, emphasizing exclusivity and luxury could be important. I should also mention the target audience. The text might talk about catering to discerning customers who want top-tier products. Maybe add something about setting a new benchmark in the industry. Including a quote from a company executive would add credibility. The name of the executive, like CEO or MD, and their comments on market growth and customer expectations would be good. Finally, wrap it up by stating the company's commitment to innovation and quality. Maybe mention future plans for expanding the product line. I need to make sure the continuation is smooth and follows the original text's tone and structure. Let me put it all together now. `</think>` |
| **Answer (Text)** | God reja Eleganza series, designed to redefine luxury in home appliances. The lineup includes smart, energy-efficient, and aesthetically crafted products such as AI-powered refrigerators, 360-degree induction cooktops, and whisper-quiet, high-capacity washing machines. Emphasizing cutting-edge technology, the Eleganza series integrates IoT-enabled features, premium stainless steel finishes, and eco-friendly manufacturing processes.Godreya Group's CEO, Adi Godreya, stated, The super premium segment is witnessing rapid growth, driven by urban consumers seeking sophisticated, reliable, and sustainable solutions. Elegenzia is not just a product line but a lifestyle statement for those who demand excellence in every aspect of their home. Priced between Rs 50,01 and Rs 3 lakh, the collection targets high-net-worth individuals and design-conscious households, aligning with India's rising aspirational market. With this launch, Godreaj aims to capture a significant share of the Rs. 11,500 crore home appliances sector, reinforcing its commitment to blending innovation with timeless elegance. |

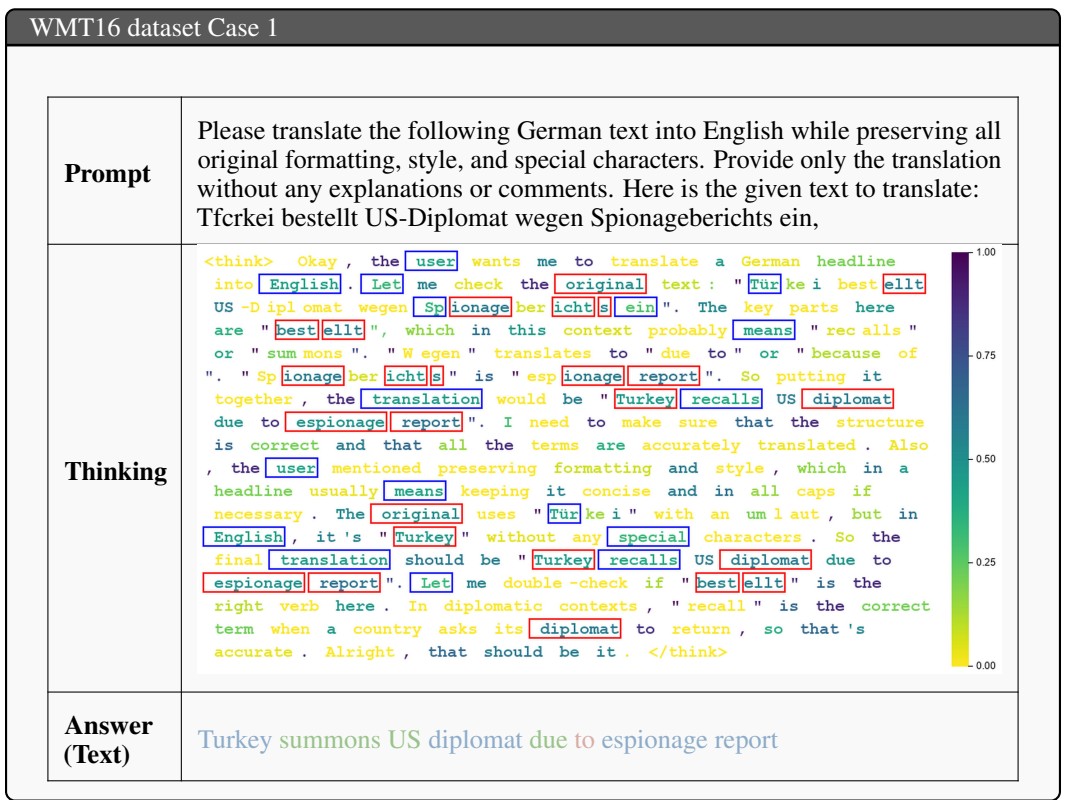

WMT16 dataset Case 1

| | |
|---|---|
| **Prompt** | Please translate the following German text into English while preserving all original formatting, style, and special characters. Provide only the translation without any explanations or comments. Here is the given text to translate: Tfcrkei bestellt US-Diplomat wegen Spionageberichts ein, |
| **Thinking** | `<think>` Okay, the user wants me to translate a German headline into English. Let me check the original text : " Türke i bestellt US -D ipl omat wegen Spionage ber icht s ein ". The key parts here are " bestellt ", which in this context probably means " rec alls " or " sum mons ". " W egen " translates to " due to " or " because of ". " Sp ionage ber icht s " is " esp ionage report ". So putting it together , the translation would be " Turkey recalls US diplomat due to espionage report ". I need to make sure that the structure is correct and that all the terms are accurately translated . Also , the user mentioned preserving formatting and style , which in a headline usually means keeping it concise and in all caps if necessary . The original uses " Türke i " with an um l aut , but in English , it 's " Turkey " without any special characters . So the final translation should be " Turkey recalls US diplomat due to espionage report ". Let me double -check if " bestellt " is the right verb here . In diplomatic contexts , " recall " is the correct term when a country asks its diplomat to return , so that 's accurate . Alright , that should be it . `</think>` |
| **Answer (Text)** | Turkey summons US diplomat due to espionage report |

## WMT16 dataset Case 2

| | |
|---|---|
| **Prompt** | Please translate the following German text into English while preserving all original formatting, style, and special characters. Provide only the translation without any explanations or comments. Here is the given text to translate: Bis dahin gab es in Kollbach im Schulverband Petershausen-Kollbach drei Klassen und in Petershausen ffcnf., |
| **Thinking** | 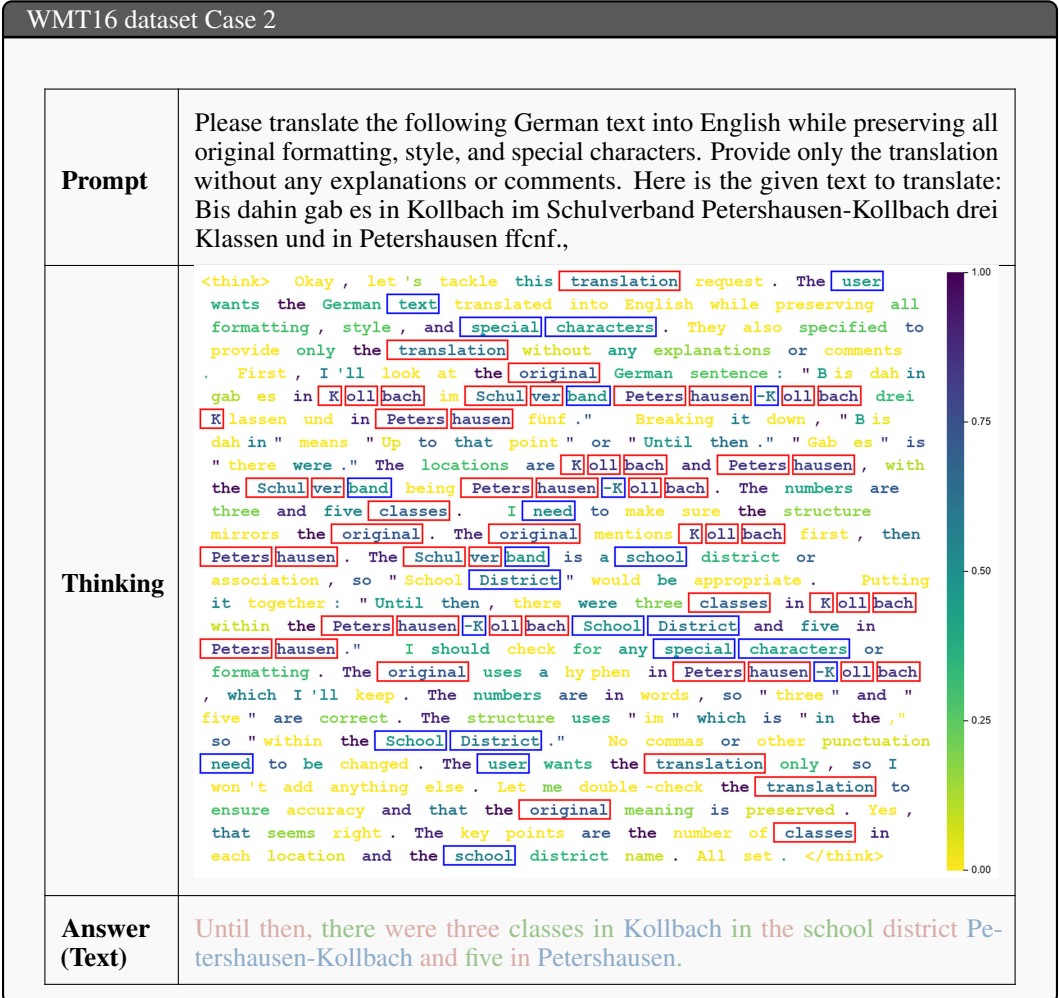 |
| **Answer (Text)** | Until then, there were three classes in Kollbach in the school district Petershausen-Kollbach and five in Petershausen. |

**GSM8K dataset Case 1**

| | |
|---|---|
| **Prompt** | Please reason step by step, and put your final answer within \boxed{}. Here is the problem: Mariah 2019s grandma was teaching her to knit. Mariah used 1/4 of a skein of yarn. Her grandma used 1/2 of a skein of yarn. There are 364 yards in a skein of yarn. How many yards of yarn did they use altogether? |
| **Thinking** | \<think\> Okay, let me try to figure out how many yards of yarn Mariah and her grandma used altogether. Hmm, so Mariah used 1/4 of a skein, and her grandma used 1/2 of a skein. Each skein has 364 yards. Alright, so I need to find out how many yards each of them used individually and then add them together, right? First, let me think about Mariah's part. If one full skein is 364 yards, then 1/4 of that would be ... Well, dividing 364 by 4. Let me do that calculation. 364 divided by 4. Let's see, 4 times 90 is 360, so 90 with a remainder of 4, which is 1 more. So 90 + 1 = 91. Wait, is that right? Let me check again. 4 times 90 is 360, and 364 minus 360 is 4, so 4 divided by 4 is 1. So yes, 91 yards. So Mariah used 91 yards. Now for her grandma, who used 1/2 of a skein. So half of 364 yards. To find half of 364, I can divide 364 by 2. Let's calculate that. 2 times 180 is 360, so 180 with a remainder of 4. Half of 4 is 2, so 180 + 2 = 182. Therefore, her grandma used 182 yards. Now, to find the total yards they used together, I need to add Mariah's 91 yards and her grandma's 182 yards. Let me add them up. 91 + 182. Let's break it down: 90 + 180 is 270, and then 1 + 2 is 3. So 270 + 3 = 273. So altogether, they used 273 yards of yarn. Wait, let me verify if there's another way to do this. Instead of calculating each person's yards separately, maybe I can add the fractions first and then multiply by the total yards per skein. Let me try that method to check if I get the same answer. So Mariah used 1/4 and her grandma used 1/2. Adding those fractions together: 1/4 + 1/2. To add them, they need a common denominator. The common denominator for 4 and 2 is 4. So 1/2 is equivalent to 2/4. Therefore, 1/4 + 2/4 = 3/4. So altogether, they used 3/4 of a skein. Then, multiplying 3/4 by 364 yards. Let me compute that. First, divide 364 by 4, which we already did earlier and got 91. Then multiply by 3: 91 * 3. Let me calculate 90*3 + 1*3 = 270 + 3 = 273. Same result! Okay, so that confirms that the total is indeed 273 yards. So both methods lead me to the same answer, which is reassuring. I think that's solid. Let me just recap to make sure I didn't miss anything. The problem asked for the total yards used by both Mariah and her grandma. Each used a fraction of a skein, and each skein is 364 yards. Calculating each fraction's yards and adding them gives the same result as adding the fractions first and then calculating the total. Both ways, it 's 273. I don't see any mistakes in my calculations. Dividing 364 by 4 gives 91, half of 364 is 182, adding them gives 273. Alternatively, adding 1/4 and 1/2 gives 3/4, three-fourths of 364 is 273. Yep, that checks out. So I think that's the correct answer. ** Final Answer ** They used a total of \boxed{273} yards of yarn. \</think\> |
| **Answer (Text)** | To determine how many yards of yarn Mariah and her grandma used together, we begin by analyzing the fraction of the skein used and then convert that into yards using the conversion factor provided. Step 1 Understand the Conversion Factor. We are given that: Each skein of yarn contains 364 yards, Mariah used $\frac{1}{4}$ of a skein, Her grandma used $\frac{1}{2}$ of a skein. Step 2 Calculate Yards Used by Each Person. Mariah's usage: $\frac{1}{4} \times 364 = 91$ yards, Grandma's usage: $\frac{1}{2} \times 364 = 182$ yards. Step 3 Add the Amounts Used by Both: $91 + 182 = 273$ yards. Step 4 Verify the Total Using Fractional Addition: $\frac{1}{4} + \frac{1}{2} = \frac{3}{4}, \frac{3}{4} \times 364 = 273$ yards. This confirms our earlier result. Final Answer: $\boxed{273}$ |

GSM8K dataset Case 2

| | |
|---|---|
| **Prompt** | Please reason step by step, and put your final answer within \boxed. Here is the problem: James decides to buy birthday candles for his 2 sons. One of them is 12 and the other is 4 years younger. A pack of 5 candles costs $3. How much does James spend on candles? |
| **Thinking** | 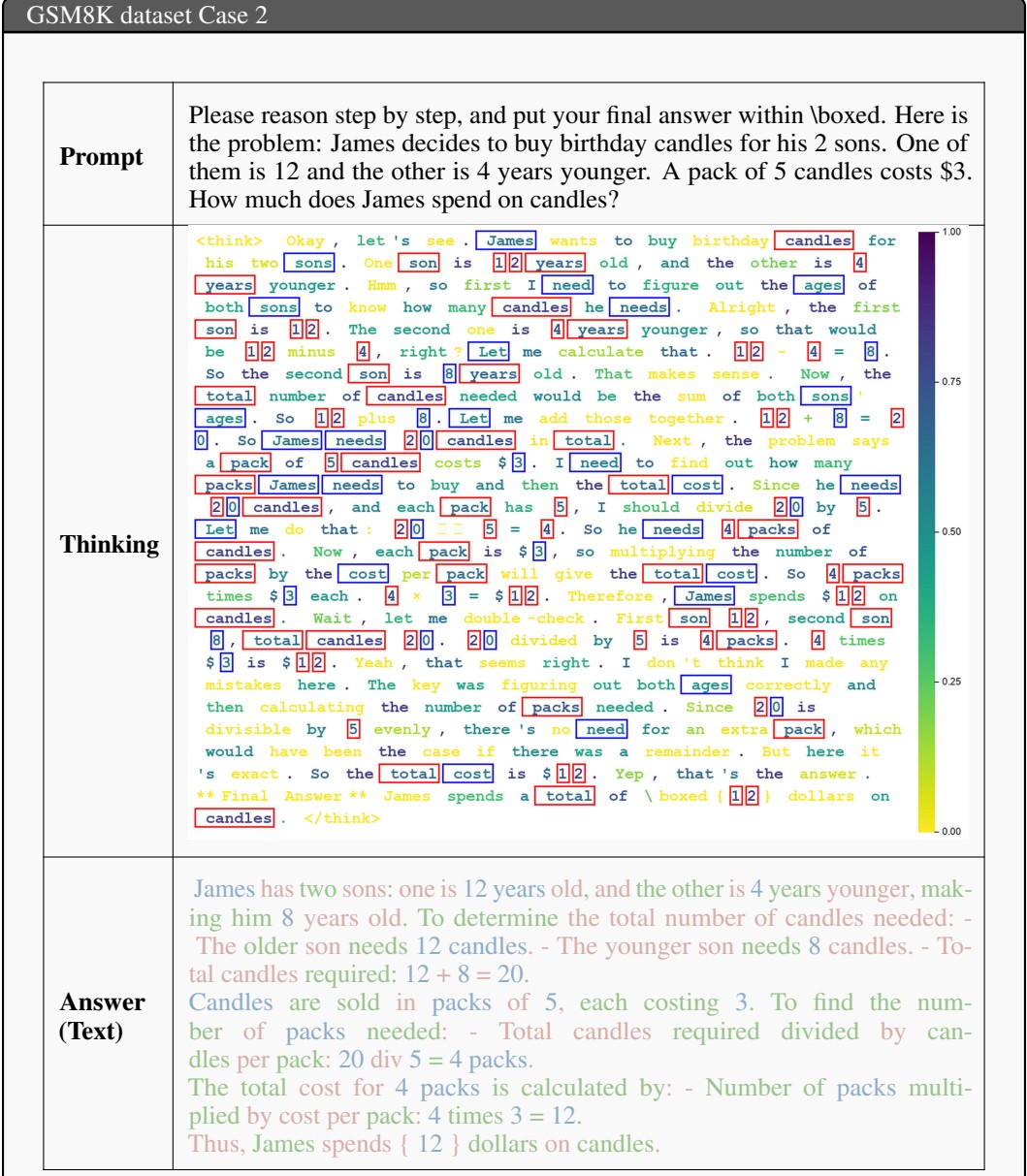 |
| **Answer (Text)** | James has two sons: one is 12 years old, and the other is 4 years younger, making him 8 years old. To determine the total number of candles needed: - The older son needs 12 candles. - The younger son needs 8 candles. - Total candles required: 12 + 8 = 20.

Candles are sold in packs of 5, each costing 3. To find the number of packs needed: - Total candles required divided by candles per pack: 20 div 5 = 4 packs.

The total cost for 4 packs is calculated by: - Number of packs multiplied by cost per pack: 4 times 3 = 12.

Thus, James spends { 12 } dollars on candles. |

