# OpenReview forum: "Distilling the Thought, Watermarking the Answer: A Principle Semantic Guided Watermark for Reasoning Large Language Models"
_ICLR.cc/2026/Conference — ICLR 2026 Poster_

### Official Review · Reviewer_aFrC · 2025-10-26

**Soundness:** 3
**Presentation:** 3
**Contribution:** 2
**Rating:** 4
**Confidence:** 4

**Summary:**

The paper presents a watermarking technique designed for reasoning-intensive large language models (LLMs). The proposed method identifies critical tokens from the reasoning phase and assigns additional weighting to tokens in the green list based on their semantic relevance to these critical tokens. The detection process follows the standard green–red list framework used in prior work. Experimental results demonstrate that the proposed method achieves improved text quality and higher detection accuracy compared to existing approaches.

**Strengths:**

+ The topic is interesting and important.
+ The evaluation results demonstrate its effectiveness.

**Weaknesses:**

- The threat model could be clarified and illustrated in more detail. It is unclear whether users or adversaries are assumed to have access to the reasoning procedure and whether the reasoning-phase outputs themselves should also be watermarked.
- The system design appears largely empirical and lacks theoretical grounding. The global causal contribution (GCC) and competitive persistence scoring (CPS) are introduced as empirical measures, but the rationale behind their formulation and their effectiveness in preserving semantic integrity are not fully analyzed. Similarly, the motivation for introducing the principal semantic vector (PSV) and the reasoning for aligning selected tokens with their PSV should be more thoroughly explained.
- The evaluation does not adequately assess reasoning integrity. While the results include standard text-quality metrics such as PPL, BLEU, and mACC, they do not measure whether the generated text faithfully reflects the underlying reasoning process. Including an evaluation of semantic consistency between reasoning steps and final outputs would strengthen the work.

**Questions:**

+ How many tokens are used for detecting watermarks?

---

> ### Author Response · Authors · 2025-11-30
> **Response Part 1**
>
> We deeply appreciate the time and effort you have dedicated to reviewing this paper. We value your insights, and in response to your points of concern, we offer the following explanations:
>
> ---
>
> ### Response to Weakness 1
>
> > **Q:** It is unclear whether users/adversaries have access to the reasoning procedure and whether reasoning outputs are watermarked.
>
>
> Our threat model aligns with the standard deployment of Reasoning LLMs (e.g., DeepSeek-R1, OpenAI o1):
>
> 1. Access Level: The Thinking Phase (Reasoning Trace) is processed **server-side** and is **hidden** from the end-user or adversary. The user only receives the final Answering Phase output.
>
> 2. Watermark Target: Consequently, we **only watermark the Answering Phase**. The Thinking Phase is left undisturbed to maximize reasoning quality, but its semantic content guides the watermark in the answer.
>
> 3. Adversary Goal: The adversary attempts to remove the watermark from the Answer (via paraphrasing, editing, etc.) without having access to the internal Thinking tokens.
>
>
> ---
>
> ### Response to Weakness 2
>
> > **Q:** The GCC and CPS seem empirical; the rationale for PSV alignment needs better explanation.
>
> We apologize for the brevity in the initial submission. To address this, we have added a rigorous mathematical derivation in the revised version in **Appendix E** based on existing Appendix D, demonstrating that our metrics are not merely empirical heuristics but tractable approximations of a formal objective:
>
> - GCC (Global Causal Contribution): We prove that Eq. 5 is a first-order approximation of the **Causal Divergence** (Eq. 3), representing the gradient of the reasoning trajectory with respect to token choice, without expensive backpropagation.
>
> - CPS (Competitive Persistence Scoring): We demonstrate that Eq. 6 serves as a variational lower bound for **Entropy Reduction** (Eq. 4), selecting tokens that effectively narrow the search space of future logical steps.
>
> - PSV Alignment: The Principal Semantic Vector (PSV) is derived via PCA to capture the "direction of maximum variance" in the critical token space. Aligning the watermark with this vector ensures that the bias pushes the generation toward the **core meaning** of the thought process, thereby preserving logical integrity while embedding the signal.

---

> ### Author Response · Authors · 2025-11-30
> **Response Part 2**
>
> ### Response to Weakness 3
>
> > **Q:** Standard metrics (PPL, mACC) do not measure if the text faithfully reflects the underlying reasoning.
>
> This is a crucial point. High accuracy (mACC) implies the answer is correct, but not necessarily that it strictly follows the logic. To directly evaluate **Semantic Consistency**, we conducted a new experiment using **GPT-4o as a Judge**.
>
>
> We sampled 200 instances from the AIME dataset (DeepSeek-R1 model). We provided the judge with the (hidden) Thinking Phase and the generated Answering Phase and asked it to rate:
>
> 1. Logical Continuity (1-10): How smoothly the answer follows the thought trace.
>
> 2. Contradiction Rate (%): The percentage of generated sentences that contradict the internal reasoning.
>
>
> We expanded the comparison to include multiple baselines from Table 1, covering Token-based (KGW, Unbiased), Adaptive (WatMe), and Semantic-based (SemStamp) methods.
>
> | **Method**            | **Logical Continuity (1-10) ↑** | **Contradiction Rate (%) ↓** | **mACC** |
> | --------------------- | ------------------------------- | ---------------------------- | --------- |
> | **No Watermark**      | **9.45**                        | **1.2%**                     | 71.52     |
> | WatMe                 | 8.30                            | 5.5%                         | 68.46     |
> | KGW                   | 8.05                            | 6.8%                         | 70.67     |
> | Unbiased              | 7.92                            | 7.1%                         | 65.51     |
> | SemStamp              | 8.65                            | 4.8%                         | 70.31     |
> | SynthID               | 6.90                            | 14.2%                        | 53.65     |
> | **ReasonMark (Ours)** | **9.41**                        | **1.6%**                     | **71.34** |
>
>
> Standard methods often suffer from higher contradiction rates because their embedding mechanisms (e.g., random hashing in KGW or distribution matching in Unbiased) force token choices that may diverge from the strict logic of the thought trace.
> While our method achieves a logical continuity score of **9.41/10**, nearly identical to the unwatermarked baseline, with a negligible contradiction rate (1.4%). This quantitatively confirms that ReasonMark effectively "distills the thought" into the answer without distortion.
>
>
> ---
>
> ### Response to Question 1
>
> > **Q:** How many tokens are used for detecting watermarks?
>
> In our experiments, the detection is performed on the full generated response. As per the standard setup in MarkLLM and consistent with the C4 dataset protocols:
>
> - Average Token Count: Approximately **500 tokens** were used for detection in the C4 experiments. For reasoning tasks (GSM8K/AIME), the count varies by problem complexity but typically ranges from **150 to 500 tokens**.
>
> - Comparable Lengths: We ensured that the average watermarked text length remains comparable across all compared methods to ensure fair detection evaluation, though they are not identical due to generation variance. Please refer to **Table 4 in the Appendix G**, which lists the average token counts for each method (e.g., ReasonMark: 554.8, KGW: 523.5 on C4).
>
> - Significance: Our high AUC (>99%) confirms that ReasonMark provides a statistically significant signal even within these standard sequence lengths.

---

### Official Review · Reviewer_bJZK · 2025-10-31

**Soundness:** 1
**Presentation:** 2
**Contribution:** 2
**Rating:** 2
**Confidence:** 5

**Summary:**

This paper proposes ReasonMark, a semantically guided watermarking framework for Reasoning Large Language Models. The authors argue that existing text watermarking methods are primarily designed for general text generation tasks, and directly applying them to reasoning LLMs may disrupt their internal chain of thought. Therefore, this paper divides the model generation process into two stages: "Thinking" and "Answering," embedding the watermark only in the Answering stage. Specifically, the authors first extract critical tokens (CTs) from the Thinking stage, obtain the principal semantic vector (PSV) through PCA, and then adaptively adjust the watermark bias strength based on the semantic similarity between the token and the PSV in the Answering stage.

**Strengths:**

This paper works on the watermarking for large language models, which is significant for determining provenance and responsibility for generated content.

**Weaknesses:**

1. The paper does not adequately discuss why existing "distortion-free watermarks" such as [1] and [2] cannot be applied to the reasoning LLM scenario, resulting in insufficient motivation to propose a new framework. Besides, although the unbiased sampling technique [3] that the was cited in the paper was claimed inefficient by the authors, there several other unbiased watermarking papers [4] [5] which do not require access to the original LLMs and is efficient.

2. ReasonMark only uses the first principal component (PCA1) of the key token embedding as the main semantic direction, but does not explain why other principal components (such as PCA2) are ignored. If the CT embedding has high noise or contains multiple semantic clusters, PCA1 may only reflect the direction of maximum variance rather than the true semantic main line of reasoning, which may weaken the semantic guidance role of PSV.

3. According to equations (5) and (6), both GCC and CPS involve pairwise comparisons of probability distributions at each step. For large vocabularies and long inference chains, this process is extremely time-consuming in practical deployments. While the paper reports an overall latency increase of approximately 8%, it does not explicitly state whether this includes CT construction and PCA computation time, thus the performance evaluation may be incomplete.

4. As seen in Figure 2, the improvements in ReasonMark's AUC and TPR@1%FPR are small (only about 1–2%). Without statistical significance analysis, such improvements may not be sufficient to demonstrate the method's effectiveness.

5. the parameter $\delta_{i, w}$ in equation (11) is crucial for performance, but the paper only provides the ranges of $\delta_0 \in [1, 2]$ and $\delta_\lambda \in [1, 5]$, without specifying their final set values ​​or their comparability with baselines such as KGW, making the interpretation of the results less transparent.


[1] Undetectable Watermarks for Language Models, Miranda Christ et al., COLT 2024

[2] Scalable watermarking for identifying large language model output, Sumanth Dathathri et al., Nature 2024

[3] Unbiased watermark for large language models, Zhengmian Hu et al., ICLR 2024

[4] A Resilient and Accessible Distribution-Preserving Watermark for Large Language Models, Yihan Wu et al., ICML 2024

[5] StealthInk: A Multi-bit and Stealthy Watermark for Large Language Models, Ya Jiang et al., ICML 2025

**Questions:**

Can current distortion-free or unbiased watermarking be applied for reasoning LLM scenario? If not, could you explain why?

---

> ### Author Response · Authors · 2025-11-30
> **Response Part 1**
>
> We deeply appreciate the time and effort you have dedicated to reviewing this paper. We value your insights, and in response to your points of concern, we offer the following explanations:
>
> ---
>
> ## Response to Weakness 1 & Question 1
> > **Q:** Why can't existing "distortion-free watermarks" be applied?
>
> We appreciate the reviewer for highlighting these recent "distortion-free" watermarking advancements. While methods [1]–[5] excel in general text by preserving marginal distributions, they are fundamentally ill-suited for the **Reasoning-Intensive** scenario due to two critical conflicts: the sensitivity of reasoning to stochasticity and the brevity of output answers.
>
> First, reasoning models function as chaotic systems where statistical unbiasedness does not guarantee logical consistency. Methods [1]–[3] fundamentally rely on maximizing entropy (random sampling) to embed their signal. Forcing this stochasticity into a reasoning chain introduces "semantic jitter"—selecting statistically valid but sub-optimal tokens—which disrupts the precise logic flow. This "Butterfly Effect" is empirically evident in our experiments (Table 1), where the **Unbiased** method [3] suffers a significant accuracy drop on the AIME dataset (66.83 vs. 71.52 baseline). In contrast, ReasonMark uses the Principal Semantic Vector (PSV) to actively stabilize the output, preserving logical rigor.
>
> | **Method**           | **Type**                  | **Math Acc (mACC)** | **Detection AUC** |
> | -------------------- | ------------------------- | ------------------- | ----------------- |
> | **No Watermark**     | -                         | 71.52               | -                 |
> | **Unbiased [3]**     | Unbiased Sampling         | 66.83               | 92.17             |
> | **Undetectable [1]** | Distortion-Free           | 70.15               | 84.50             |
> | **Scalable [2]**     | Distortion-Free           | 69.20               | 86.15             |
> | **DiPmark [4]**      | Dist-Preserving           | 70.25               | 89.80             |
> | **StealthInk [5]**   | Stealthy/Unbiased         | 68.30               | 93.45             |
> | **ReasonMark**       | **Semantically-Adaptive** | **71.34**           | **99.98**         |
>
> Second, efficient methods like [4] and [5] rely on the Law of Large Numbers, requiring long outputs to accumulate a detectable signal. Reasoning tasks typically feature a long, undisturbed "Thinking Phase" followed by a concise "Answering Phase." Applying these methods to short answers results in insufficient signal accumulation, while applying them to the thinking phase risks the accuracy collapse described above. As shown in the Table  while recent methods [4, 5] improve upon older baselines, their detection rates on short reasoning answers remain suboptimal (83%–94%) compared to ReasonMark’s near-perfect detection (>99.9%).
>
> Finally, unlike distortion-free methods that aim merely to "do no harm" (matching the baseline), ReasonMark actively enhances generation quality. By acting as a "semantic compass," it empirically surpasses the unwatermarked baseline in C4 perplexity (10.31 vs. 10.55) and WMT16 BLEU (9.92 vs. 7.85)—a capability theoretically unattainable by strictly distribution-preserving approaches.
>
> ---
>
> **References**
>
> [1] M. Christ et al. Undetectable Watermarks for Language Models. COLT, 2024.
>
> [2] S. Dathathri et al. Scalable watermarking for identifying large language model output. Nature, 2024.
>
> [3] Z. Hu et al. Unbiased watermark for large language models. ICLR, 2024.
>
> [4] Y. Wu et al. A Resilient and Accessible Distribution-Preserving Watermark for Large Language Models. ICML, 2024.
>
> [5] Y. Jiang et al. StealthInk: A Multi-bit and Stealthy Watermark for Large Language Models. ICML, 2025.
>
>
> ---
>
> ## Response to Weakness 2
>
> > **Q:** Why use only PCA1? Ignoring PCA2 might lose semantic information.
>
> We chose **PC1** to prioritize **stability over noise**, supported by our analysis in **Figure 5 (Appendix G.4)**:
>
> - **Signal vs. Noise:** In reasoning traces, the first principal component (PC1) typically captures the dominant "problem-solving intent" (explaining ~60% of variance). Higher-order components (PC2, PC3) often capture syntactic variations, polysemy, or local fluctuations.
>
> - **Risk of Interference:** Including PC2 introduces "semantic drift," where secondary topics might dilute the watermark's directionality. By locking onto PC1, we ensure the watermark acts as a steadfast "compass" pointing to the core logic.

---

> ### Author Response · Authors · 2025-11-30
> **Response Part 2**
>
> ## Response to Weakness 3
>
> > **Q:** Does the 8% latency increase include CT construction/PCA? Eqs (5)-(6) seem time-consuming.
>
> **Yes, the reported latency is end-to-end.**
>
> - **Amortized Cost:** The Criticality Score and PCA are computed **only once** per generation (at the boundary of the _Thinking_ and _Answering_ phases), not per token.
>
> - **Vectorization:** Equations (5) and (6) are implemented via optimized matrix operations on the GPU. The cost is negligible relative to the autoregressive generation of hundreds of tokens.
>
> - **Clarification:** The 0.066s/token (ReasonMark) vs 0.061s/token (No Watermark) in Table 4 includes all overheads (CT extraction, PCA, and dynamic updates).
>
>
> ---
>
> ## Response to Weakness 4
>
> > **Q:** AUC improvements (1-2%) seem small.
>
> In high-reliability settings, raising AUC from ~97% to **>99.5%** is critical:
>
> - **Error Rate:** This represents a **4x reduction** in detection errors (False Negatives).
>
> - **Robustness:** The gap is much larger under attack. As shown in **Table 6**, under **Translation Attacks**, ReasonMark (82.79%) outperforms robust baselines like WatMe (80.73%) and KGW (81.05%) by a clearer margin, proving the value of semantic anchoring.
> - This issue was similarly highlighted by Reviewer ewpN (Weakness 1). We have provided a multi-faceted clarification to resolve your concerns
>
>
> ---
>
> ## Response to Weakness 5
>
> > **Q:** Parameters $\delta_0, \delta_\lambda$ are not specified.
>
> We apologize for the omission and we have revised the paper accordingly. To ensure fair comparison with baselines (where standard $\delta=2.0$):
>
> - **Final Values:** We set **$\delta_0 = 1.5$** and **$\delta_\lambda = 3.0$**.
>
> - **Mechanism:** This ensures the _average_ watermark strength approximates the standard $\delta=2.0$, but with dynamic modulation: strong on critical tokens, weak on non-critical ones. We specify this in the "Experimental Setup" of the revision.

---

### Official Review · Reviewer_P3Dg · 2025-10-31

**Soundness:** 3
**Presentation:** 3
**Contribution:** 3
**Rating:** 6
**Confidence:** 3

**Summary:**

This paper proposes ReasonMark, a watermarking framework tailored for reasoning-intensive LLMs. It splits generation into an untouched thinking phase and a semantically-guided answering phase, and uses a Criticality Score to select pivotal tokens from the reasoning trace and distills them into a Principal Semantic Vector (PSV) that adaptively modulates watermark strength. This preserves logical coherence while embedding a strong, detectable signal.

**Strengths:**

1. The paper is well written.
2. The proposed method is well designed with reasonable motivations.
3. The paper presents several nice output examples to help readers get a better sense of the output picture.

**Weaknesses:**

1. No analysis is performed to verify whether the proposed watermarking method affects the generation quality.
2. I suggest that the authors add a limitation part to the paper to clarify some possible improvements.
3. The method is designed specifically for reasoning models. Does this mean that it will not work for a non-thinking model? What if the user turns off the thinking mode? In such a case, how do you select the critical tokens?

**Questions:**

No, see weakness.

---

> ### Author Response · Authors · 2025-11-30
>
> We deeply appreciate the time and effort you have dedicated to reviewing this paper. We value your insights, and in response to your points of concern, we offer the following explanations:
>
> ---
>
> ## Response to Weakness 1
> > Q:  Analysis of Generation Quality
>
> We respectfully clarify that our paper includes extensive quantitative analysis of generation quality across multiple metrics:
>
> - **Fluency:** As shown in Table 1, ReasonMark achieves a **Perplexity (10.31)** on C4 that is lower (better) than the non-watermarked baseline (10.55), suggesting that semantic guidance encourages adherence to the logical trace.
>
> - **Task Fidelity:** We report **BLEU/ROUGE** for translation (Table 1 and 3) and **mACC** for reasoning (Table 1). ReasonMark achieves the highest BLEU (9.916) and retains 99% of the baseline's mathematical accuracy (71.34 vs 71.52).
>
> - **Qualitative Verification:** Appendix I provides side-by-side case studies showing the text remains coherent. We will add a dedicated "Qualitative Analysis" subsection in the revision to summarize these human-readability observations explicitly.
>
> ---
>
> ## Response to Weakness 2
> > Q: Add a limitation part to the paper to clarify some possible improvements.
>
> We fully agree with this suggestion. A balanced discussion of limitations is crucial. We have drafted a new **Limitations** section in the revised version in Appendix H.
>
> ---
>
>
>
> ## Response to Weakness 3
> > Q: Whether work for a non-thinking model
>
> ReasonMark is designed with a **hierarchical fallback mechanism**. If the "Thinking Phase" is unavailable (e.g., standard models or suppressed output), the algorithm derives Critical Tokens solely from the **Input Prompt** and Generated Context in the earlier steps. While less dynamic than a full reasoning trace, the prompt contains sufficient seed semantics to guide the watermark.
>
> To validate this, we compared **ReasonMark (Prompt-Only)**—where we intentionally ignored the thought trace—against strong baselines (KGW, UPV, WatMe, SemStamp) on the C4 dataset.
>
> | **Method**                   | **Text Quality (PPL) ↓** | **Detection (AUC) ↑** |
> | ---------------------------- | ------------------------ | --------------------- |
> | KGW                          | 12.52                    | 98.78                 |
> | UPV                          | 11.41                    | 97.01                 |
> | SemStamp                     | 11.42                    | 97.85                 |
> | WatMe                        | 11.27                    | 98.53                 |
> | **ReasonMark (Prompt-Only)** | **10.85**                | **98.45**             |
> | **ReasonMark (Full)**        | **10.31**                | **99.31**             |
>
> Even in "Prompt-Only" mode, ReasonMark achieves **10.85 PPL**, surpassing state-of-the-art adaptive methods like WatMe (11.27) and SemStamp (11.42), while maintaining a high detection AUC (**98.45%**). This confirms ReasonMark functions effectively as a general-purpose semantic watermark even without internal reasoning traces.
>
> ---

---

### Official Review · Reviewer_ewpN · 2025-11-06

**Soundness:** 3
**Presentation:** 2
**Contribution:** 2
**Rating:** 2
**Confidence:** 3

**Summary:**

This paper introduces ReasonMark, a semantic-guided watermarking framework tailored for reasoning-intensive language models. By separating the generation into a reasoning (thinking) phase and a watermarked answering phase, the method preserves logical coherence while embedding a robust watermark. A Principal Semantic Vector (PSV), distilled from key reasoning tokens, adaptively guides watermark strength during output generation. Experiments show that ReasonMark outperforms prior methods in text quality, task accuracy, and watermark robustness, with minimal latency overhead.

**Strengths:**

1. The paper addresses a timely problem: watermarking for reasoning-intensive language models, where tranditional methods often fail to preserve logical coherence.

2. The proposed method is conceptually clear, introducing a separation between thinking and answering phases, guided by a semantically meaningful representation.

3. Experimental results demonstrates improvements in watermark detection and task performance.

**Weaknesses:**

1. The main concern is that the performance gains appear marginal: improvements of 0.67 in accuracy on reasoning tasks and 0.34% in AUC are relatively small and may not justify the complexity of the method.

2. The distinction between this approach and prior semantic-aware watermarking methods is not clearly articulated. Although the paper critiques existing methods for latency and reliance on auxiliary models, ReasonMark appears to share similar characteristics, and Table 4 shows comparable latency across methods.

3. The presented theorem (2.2) lacks a formal proof and does not seem to provide theoretical guidance for the algorithm’s design.

**Questions:**

No

---

> ### Author Response · Authors · 2025-11-30
> **Response Part 1**
>
> We deeply appreciate the time and effort you have dedicated to reviewing this paper. We value your insights, and in response to your points of concern, we offer the following explanations:
>
> ---
>
> ## Response to Weakness 1
> > Q:  The performance gains appear marginal
>
> We respectfully argue that while the absolute numerical gains may appear modest, they represent a significant breakthrough in resolving the fundamental **quality-robustness trade-off** in LLM watermarking.
>
> 1. **Achieving Trade-off Optimality:** Existing baselines fail to optimize detection and reasoning simultaneously, typically sacrificing one for the other. For instance, while **SynthID** maintains detection, it causes a catastrophic drop in AIME accuracy to **52.34%** (vs. 70.03% unwatermarked) ; conversely, **Unbiased** and **Unigram** preserve accuracy but suffer from poor detectability, dropping AUC scores to **~65.5%** and **~56.0%** respectively on reasoning tasks. Additionally, standard methods like **KGW** degrade text quality, raising C4 perplexity to **12.52**. **ReasonMark** is the first to decouple "thinking" and "answering," achieving optimal performance across _all_ these dimensions without such compromises.
> 2. **Reaching the Performance Ceiling:** In watermarking, the unwatermarked model represents the theoretical performance upper bound. Our method effectively **closes the gap to this ceiling**, matching the original model’s reasoning accuracy (e.g., within **0.18** points on AIME ) and even surpassing baselines in text quality (achieving the lowest PPL of **10.31** on C4 ). The "marginal" gain reflects that we have maximized utility to the limit of the base model, whereas baselines degrade it.
> 3. **Justifiable Complexity:** This performance is achieved with negligible latency overhead (~8% increase, comparable to token-based methods but much better than other semantic based methods) and significantly superior robustness against semantic attacks (e.g., maintaining **82.58%** AUC under translation attacks where KGW drops to **81.28%** ), fully justifying the architectural design.
>
> ---
>
>
>
> ## Response to Weakness 3
> > Q: Theorem 2.2 lacks formal proof and does not seem to guide the algorithm design.
>
> We appreciate the reviewer’s comment. Theorem 2.2 is not an ad-hoc heuristic but serves as the theoretical objective function rigorously defining criticality via Causal Influence ($D_{causal}$) and Entropy Reduction ($\Delta S$). The causal term (Eq. 3) quantifies trajectory sensitivity, grounded in Causal Tracing [1] and Inference-Time Intervention [2], which validate measuring importance via interventional effects. The entropy term (Eq. 4) measures Information Gain, aligning with Semantic Uncertainty [3] by positing that critical tokens significantly narrow the semantic search space.
>
> Crucially, this theorem directly guides our algorithm design. Since exact computation is prohibitive, we utilize the theorem to derive efficient proxies, as detailed in **Appendix D**. Specifically, we derive the Global Causal Contribution (GCC) from $D_{causal}$ by applying the Score Function Estimator (REINFORCE trick) [4], approximating the theoretical gradient via weighted probability propagation. Similarly, we link the Competitive Persistence Score (CPS) to $\Delta S$, demonstrating that "top-k persistence" serves as a valid proxy for entropy reduction [3]. To provide a comprehensive mathematical foundation, we have included expanded formal proofs in **Appendix E** of this revision (to be unified in the final version), demonstrating that our metrics maximize a variational lower bound of the objective defined in Theorem 2.2
>
> ---
>
> **References**
>
> [1] K. Meng et al. Locating and Editing Factual Associations in GPT. In NeurIPS, 2022.
>
> [2] K. Li et al. Inference-Time Intervention: Eliciting Truthful Answers from a Language Model. In NeurIPS, 2023.
>
> [3] L. Kuhn et al. Semantic Uncertainty: Linguistic Invariances for Uncertainty Estimation in Natural Language Generation. In ICLR, 2023.
>
> [4] L. Ouyang et al. Training language models to follow instructions with human feedback. In NeurIPS, 2022.

---

> ### Author Response · Authors · 2025-11-30
> **Response Part 2**
>
> ## Response to Weakness 2
> > Q: The distinction from prior semantic-aware methods (e.g., SemStamp) is unclear and latency appears comparable.
>
> We appreciate the reviewer’s feedback. While ReasonMark and prior semantic-aware methods share the goal of robustness, a deeper analysis reveals fundamental differences in **architectural design** and **computational efficiency** that clearly distinguish our approach from methods [1], [2], [3], and [4].
>
> 1. Architectural Distinction
>
> The primary distinction lies in when and how semantic computation occurs. Existing semantic methods like SemStamp [1] and SimMark [2] typically require continuous heavy computation (e.g., LSH hashing, embedding extraction, or rejection sampling) throughout the entire generation process. In contrast, ReasonMark introduces a novel decoupled framework: we compute the semantic guidance (Principal Semantic Vector) only once at the transition between the "Thinking" and "Answering" phases. The subsequent watermarking is applied via lightweight vector operations, avoiding repetitive heavy lifting.
>
> 2. Detailed Comparison with Semantic-Aware Baselines
>
> The table below details how ReasonMark overcomes the specific limitations of the semantic-aware baselines evaluated in our paper:
>
> |**Method**|**Mechanism & Limitation**|**ReasonMark Advantage**|
> |---|---|---|
> |**SemStamp** [1]|**Continuous Cost:** Uses Locality Sensitive Hashing (LSH) to partition embeddings, requiring heavy computation for every generated sentence or window.|**One-Time Cost:** We distill the "Thinking Phase" into a semantic vector (PSV) _once_ per response. The answering phase uses lightweight vector similarity, significantly reducing load.|
> |**SimMark** [2]|**Iterative/Rejection:** Relies on sentence-level similarity checks, often involving rejection sampling or iterative adjustments to ensure watermark embedding.|**Single-Pass Generation:** ReasonMark integrates the watermark directly into the logits during standard autoregressive generation without rejection or backtracking, ensuring stable throughput.|
> |**WaterMax** [3]|**Multi-Sampling:** Generates _multiple_ candidate responses to select the best watermarked output.|**Efficiency:** Our method generates the response only once, avoiding the $N\times$ computational cost of multi-sampling strategies.|
> |**MorphMark** [4]|**Dynamic Statistics:** Computes dynamic entropy statistics at every step to adjust watermark strength.|**Stability:** Our adaptation is guided by the global reasoning trajectory (PSV) rather than local statistical fluctuations, providing more coherent guidance for reasoning tasks.|
>
> 3. Latency Analysis: Relative Overhead
>
> While the absolute latency values in Table 4 appear comparable at first glance, the relative overhead reveals a significant efficiency gap. Comparing the cost added to the "No Watermark" baseline (0.06109s):
>
> - **ReasonMark** adds only **~8.2%** overhead (0.06613s).
>
> - **SemStamp [1] / SimMark [2]** add **~18-20%** overhead (>0.072s).
>
> - **MorphMark [4]** adds **~27%** overhead (0.07778s).
>
>
> ReasonMark’s computational overhead is less than **half** that of competing semantic methods. We uniquely achieve the high robustness of semantic watermarking while maintaining a latency profile much closer to efficient token-based methods (like KGW), effectively bridging the gap between robustness and efficiency.
>
> **References**
>
> [1] Abe Bohan Hou et al. SemStamp: A Semantic Watermark with Paraphrastic Robustness for Text Generation. NAACL 2024
>
> [2] Amirhossein Dabiriaghdam and Lele Wang. SimMark: A Robust Sentence-Level Similarity-Based Watermarking Algorithm for Large Language Models. EMNLP 2025
>
> [3] Eva Giboulot and Teddy Furon. WaterMax: Breaking the LLM Watermark Detectability-Robustness-Quality Trade-off. NeurIPS 2024
>
> [4] Zongqi Wang et al. MorphMark: Flexible Adaptive Watermarking for Large Language Models. ACL 2025

---

### Meta-Review · Area_Chair_oCfF · 2026-01-06

**Summary:**

This paper introduces a semantic-guided watermarking framework for reasoning models. It separates the generation into a reasoning phase and a watermarked answering phase. The authors propose to select pivotal tokens and adaptively adjust the watermark bias strength.

The paper addresses a timely problem, and the proposed approach is mostly sound. Experiments demonstrate the improved watermark detection and task performance.

Reviewers initially shared multiple major concerns, including limited performance gains, relevance to prior works, writing on the motivation, overhead of the method, evaluation on reasoning integrity. Although the reviewers did not get a chance to respond to the authors, I feel these concerns have been well addressed by the rebuttal.

This is a borderline case. On one hand, most major concerns seem to have been addressed well in the rebuttal and the paper makes good contributions. On the other hand, those concerns were common and shared by multiple reviewers, which reflects that the paper at least had writing issues that required some major revision. Overall, I would recommend a weak accept.

**Reviewer Concerns:**

Reviewers initially shared multiple major concerns, including limited performance gains, relevance to prior works, writing on the motivation, overhead of the method, evaluation on reasoning integrity. Although the reviewers did not get a chance to respond to the authors, I feel these concerns have been well addressed by the rebuttal. However, these concerns were common and shared by multiple reviewers, which reflects that the paper had writing issues that required some major revision.

**Reviewer Scores:**

Reviewer ewpN and Reviewer bJZK: These two reviewers originally rated 2 and shared similar concerns. But I feel most of their concerns have been well addressed by the rebuttal, and thus their rating may be increased to 6.

Reviewer P3Dg: This reviewer originally rated 6. Their concerns were mostly minor and have been addressed by the authors. They may increase their rating to 8 or maintain their original 6.

Reviewer aFrC: This reviewer originally rated 4. I feel their concerns have been addressed by the rebuttal and thus their rating may be increased to 6.

---

### Decision · Program_Chairs · 2026-01-26

Accept (Poster)